# DNMT1 loss leads to hypermethylation of a subset of late replicating domains by DNMT3A

Ioannis Kafetzopoulos[1,2], Francesca Taglini[1], Moira Pasquier[3],
Hazel Davidson-Smith[1], Christine J. Rodger[3], Lucia Puchades Gimeno[3],
Andrew A. Malcolm[1], Duncan Sproul[1]*

1 Institute of Genetics and Cancer, University of Edinburgh, Edinburgh, United Kingdom, 2 Current address: Altos Labs, Cambridge Institute, Cambridge, United Kingdom, 3 MRC Human Genetics Unit, Institute of Genetics and Cancer, University of Edinburgh, Edinburgh, United Kingdom

* d.sproul@ed.ac.uk

## Abstract

Loss of DNA methylation is a hallmark of cancer that is proposed to promote carcinogenesis through gene expression alterations, retrotransposon activation and induction of genomic instability. Cancer-associated hypomethylation does not occur across the whole genome but leads to the formation of partially methylated domains (PMDs). However, the mechanisms underpinning PMD formation remain unclear. PMDs replicate late in S-phase leading to the hypothesis that they become hypomethylated due to incomplete re-methylation by the maintenance methyltransferase DNMT1 during cell division. Here we investigate the role of DNMT1 in shaping the cancer methylome by conducting whole genome bisulfite sequencing (WGBS), repli-seq and ChIP-seq on DNMT1 knockout HCT116 colorectal cancer cells (DNMT1 KO cells). We find that DNMT1 loss leads to preferential hypomethylation in late replicating, heterochromatic PMDs marked by the constitutive heterochromatic mark H3K9me3 or the facultative heterochromatic mark H3K27me3. However, we also observe that a subset of H3K9me3-marked PMDs gain methylation in DNMT1 KO cells. We find that, in DNMT1 KO cells, these hypermethylated PMDs remain late replicating but DNMT3A localises to them. This is accompanied by loss of heterochromatic H3K9me3, specific gain of euchromatic H3K36me2 and some gene upregulation. These same domains also have more variable DNA methylation than other PMDs in colorectal tumours *in vivo*. Our observations suggest that hypermethylated PMDs lose their heterochromatic state, enabling their methylation by DNMT3A and the establishment of a hypermethylated, non-PMD state, despite their late replication timing. More generally, our findings suggest that differential *de novo* DNMT activity plays a key role in establishing domain level DNA methylation patterns in cancer cells.

**Data availability statement:** All sequencing data that were generated during this study are deposited in NCBI GEO series GSE297847.

**Funding:** Work in DS's laboratory was supported by a Cancer Research UK Career Development fellowship (reference C47648/A20837, provided salary for DS and FT), an MRC university grant to the MRC Human Genetics Unit and an MRC grant (reference UKRI576, provided salary for FT and part of DS's salary). IK was funded by a studentship from Cancer Research UK (C157/A25186) as well as the AG Leventis Foundation (18736). MP, CJR and LPG were supported by the MRC Human Genetics Doctoral Training Program. The funders had no role in study design, data collection and analysis, decision to publish, or preparation of the manuscript.

**Competing interests:** The authors have declared that no competing interests exist.

## Author summary

DNA methylation is a chemical modification that is thought to play a role in turning genes on and off. Cancer cells have low DNA methylation levels compared to normal cells. This could promote cancers by turning on genes at the wrong time or causing breaks in DNA. However, it is unclear if this is the case. We know that DNA methylation is not lost from all DNA, but instead from specific parts termed partially methylated domains (abbreviated to PMDs). Here we tried to understand why DNA methylation is lost specifically from PMDs by analysing cells where DNMT1, the major enzyme placing DNA methylation, was removed. Surprisingly, we saw that a few PMDs gained DNA methylation in these cells. By observing these PMDs more closely we found that they changed their packing and chemical tags. This was associated with recruitment of a different enzyme that places DNA methylation, DNMT3A. In actual colon tumours, the same regions show highly variable DNA methylation, suggesting they are especially unstable. Our study indicates that where and when different enzymes placing DNA methylation function is crucial for dictating which parts of our DNA lose DNA methylation in cancers.

## Introduction

DNA methylation is an epigenetic modification which occurs by the addition of a methyl group on the 5' position of a cytosine ring. In vertebrates, DNA methylation occurs largely in the context of CpG dinucleotides and the majority of CpGs in the genome are methylated [1,2]. DNA methylation is established in development by the de novo DNA methyltransferases (DNMTs), DNMT3A and DNMT3B [3,4]. Methylation patterns are then maintained predominantly by the maintenance DNA methyltransferase DNMT1 [5], although DNMT3A and DNMT3B also play a role [6].

DNA methylation patterns are dysregulated in cancer and overall levels of DNA methylation are reduced in tumours [7,8]. This hypomethylation is not uniform across the genome but occurs in interspersed, mega-base scale regions termed partially methylated domains (PMDs) [9–11]. Cancer-associated hypomethylation is proposed to facilitate tumorigenesis through gene expression alterations, the activation of repetitive elements and the promotion of genome instability [12–14].

PMDs display distinct genomic characteristics compared to the rest of the genome. They have a reduced mean CpG density, are gene poor and the genes within PMDs are generally repressed [1,10,15]. This suggests they are heterochromatic in nature. Indeed, PMD chromatin is resistant to nuclease digestion [16] and is marked by the repressive histone marks H3K9me3 and H3K27me3 [10,17], associated with constitutive and facultative heterochromatin respectively [18]. In support of the heterochromatic nature of PMDs, they coincide with lamina associated domains (LADs) [9,10] and overlap with the HiC defined B-compartment [17]. In addition, during S-phase, heterochromatin and PMDs are observed to replicate later than euchromatin [17,19].

The molecular mechanisms responsible for the formation of PMDs remain unclear. Pulse-chase experiments report that re-methylation of newly synthesised DNA is inefficient and can take several hours following replication [20–22]. Re-methylation kinetics are also non-uniform across the genome and heterochromatin is reported to be re-methylated more slowly than euchromatin [21,22]. This has led to the hypothesis that late replicating regions may not have sufficient time to become re-methylated following DNA replication in rapidly proliferating cancer cells [23,24]. The continuous proliferation of cancer cells could therefore lead to the hypomethylation of heterochromatin and the formation of PMDs with successive cell divisions. Indeed, PMD methylation levels negatively correlate with the number of mutations in tumours [24] suggesting that PMDs lose DNA methylation as a function of the number of cell divisions that have occurred. Given that DNMT1 is primarily responsible for maintaining DNA methylation following replication [23], this model predicts a central role for DNMT1 in the formation of PMDs. In support of this hypothesis, DNMT1 has been observed to be excluded from heterochromatic, late-replicating satellite DNA in senescent cells where extensive hypomethylation of PMDs occurs [25].

To understand how maintenance methylation in cancer cells varies with replication timing and chromatin structure, we have interrogated DNA methylation changes in DNMT1 knock-out HCT116 cells (DNMT1 KO). While we observe that late-replicating, heterochromatic regions lose more DNA methylation in DNMT1 KO cells than early replicating, euchromatic regions, we also observe that a subset of H3K9me3-marked late-replicating regions gain DNA methylation. This associates with the recruitment of DNMT3A, loss of H3K9me3 and specific gain of H3K36me2. These results therefore suggest that DNMT3A plays an important role in counteracting hypomethylation in PMDs.

## Results

### Ablation of DNMT1 leads to preferential hypomethylation of partially methylated *domains*

To understand the role of DNMT1 in shaping the cancer epigenome at the domain level, we compared the methylomes of HCT116 cells to a published DNMT1 knock-out derivative [26] using Whole Genome Bisulfite Sequencing (WGBS, HCT116 mean coverage = 1.6x, DNMT1 KO mean coverage = 2.4x). As complete removal of DNMT1 in somatic cells is lethal, these cells express a truncated DNMT1 lacking exons corresponding to the DMAP and PCNA-interacting domains, to around 20% of the level of DNMT1 in HCT116 cells [27,28]. They are therefore strictly a truncated, hypomorphic line, but for the sake of brevity we refer to these cells as DNMT1 KO cells throughout the manuscript.

Before examining the nature of the changes in DNA methylation in DNMT1 KO cells, we first assessed the DNA methylation landscape in HCT116 cells. We used the WGBS to quantify mean DNA methylation levels in 10Kb bins across the genome, finding that they were bimodally distributed due to the presence of PMDs (Fig 1A-1B). To understand the properties of these domains, we used methpipe to define 546 HCT116 PMDs on autosomal chromosomes [29]. We then defined the rest of the genome as 556 HCT116 highly methylated domains (HMDs, see methods for details).

PMDs have previously been reported to replicate late in S-phase and be marked with heterochromatic histone modifications [9,10]. To understand whether this was also the case in HCT116 cells, we performed repli-seq to measure replication timing alongside ChIP-seq for the histone modifications H3K4me3, H3K9me3, H3K27me3 and H3K36me3. We then compared replication timing between HCT116 PMDs and HMDs confirming that the replication timing of PMDs was significantly later than that of HMDs (S1A Fig, p < 2.2x10$^{-16}$, Wilcoxon rank sum test). Both the constitutive heterochromatin-associated H3K9me3 and the facultative heterochromatin-associated H3K27me3 marks had significantly higher levels in HCT116 PMDs compared to HMDs (S1B-S1C Fig, both p < 2.2x10$^{-16}$, Wilcoxon rank sum tests). H3K4me3 and H3K36me3 were observed in peaks around promoters and gene bodies in HMDs respectively (S1D Fig). However, we also noted a low, broad enrichment of both marks in PMDs (Fig 1A).

Further investigation revealed that HCT116 PMDs were split into 253 predominantly marked by H3K9me3 and 293 predominantly marked by H3K27me3 (S1E Fig). H3K9me3-marked PMDs were bordered by H3K27me3 (Figs 1A and S1E), a pattern previously reported in IMR90 and HCC1954 cells [10]. The median length of H3K9me3-enriched PMDs was

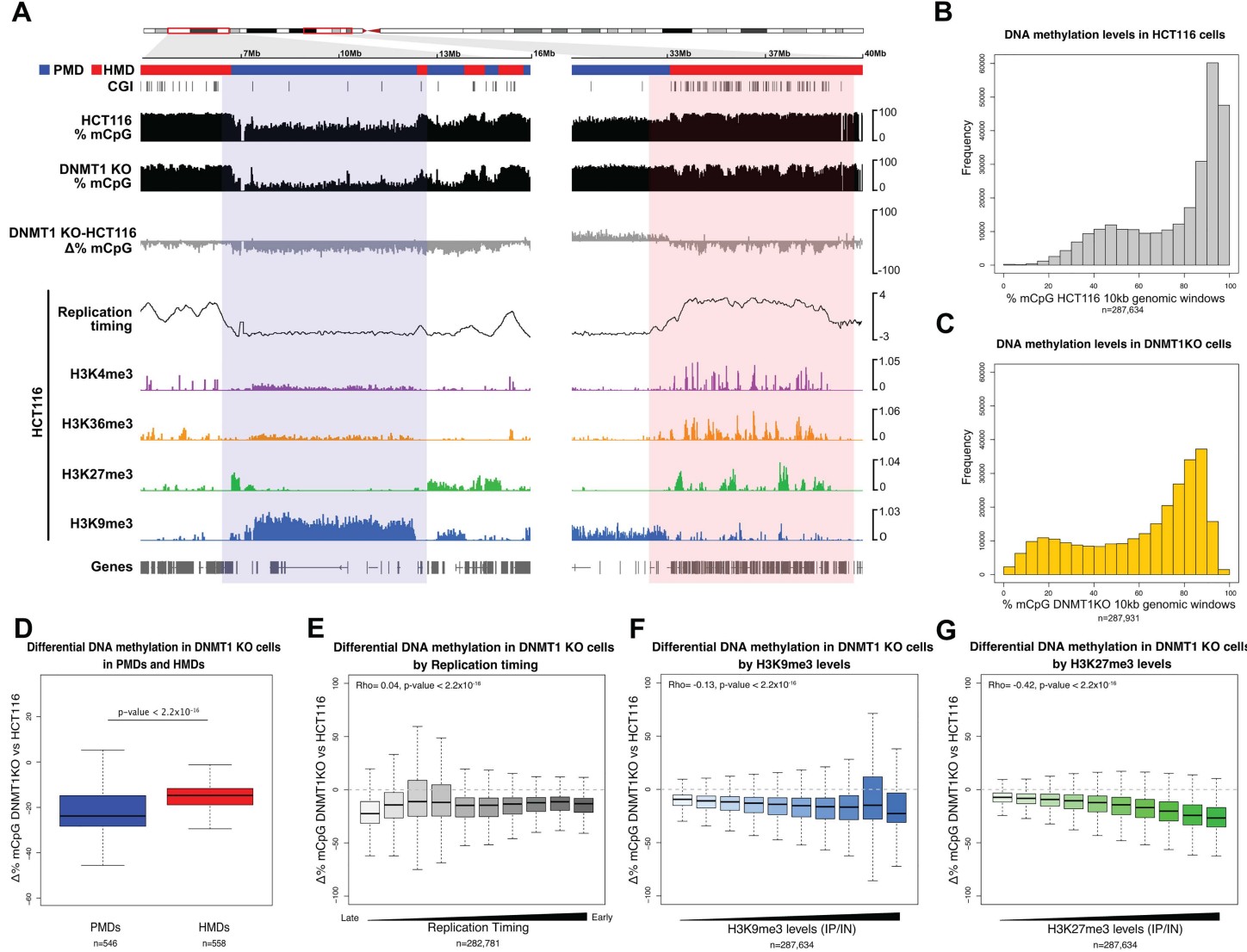

**Fig 1. Ablation of DNMT1 leads to preferential hypomethylation of partially methylated domains. (A)** Representative genomic loci showing changes of DNA methylation at a PMD and HMD in DNMT1 KO cells. Genome browser plots showing absolute (black) and differential (grey) DNA methylation levels (mCpG) alongside HCT116 histone modification ChIP-seq and repli-seq. DNA methylation levels are plotted in 10 kb genomic windows. ChIP-seq tracks are normalised $\log_{10}$ IP/IN. Replication timing data are loess smoothed repli-seq early/late ratios over 10 kb. Representative PMD and HMD are indicated by the coloured boxes. CGI = CpG island. Chromosomal co-ordinates shown are from hg38, left: chr9:4,000,000-16,000,000 and right: chr9:31,000,000-40,000,000. **(B, C)** Histograms of DNA methylation levels in HCT116 (**B**) and DNMT1 KO (**C**) cells. Mean DNA methylation levels were calculated for 10 kb genomic windows. In each case, n is the number of genomic windows included. **(D)** Boxplot showing differential DNA methylation between DNMT1 KO and HCT116 at PMDs (n = 546 domains) and HMDs (n = 558 domains) defined in HCT116 cells. P-value from two-sided Wilcoxon rank sum test. **(E)** Boxplot showing difference in DNA methylation between DNMT1 KO and HCT116 cells in 10 kb genomic windows divided in deciles according to their HCT116 replication timing. Replication timing data are loess smoothed repli-seq early/late ratios over 10 kb. **(F)** Boxplot showing difference in DNA methylation between DNMT1 KO and HCT116 cells in 10 kb genomic windows divided in deciles according to their H3K9me3 levels in HCT116 cells. ChIP-seq data are normalised IP/IN. **(G)** Boxplot showing difference in DNA methylation between DNMT1 KO and HCT116 cells in 10 kb genomic windows divided in deciles according to their H3K27me3 levels in HCT116 cells. ChIP-seq data are normalised IP/IN. For boxplots: Lines = median; box = 25th–75th percentile; whiskers = 1.5 × interquartile range from box. All histone ChIP-seq and repli-seq data shown are derived from the mean of two biological replicates. In **(E-G)**, n is the number of windows analysed shown below the plots and the Spearman's correlation coefficient (Rho) is shown alongside its associated p-value.

1.86Mb compared to 0.59Mb for H3K27me3-enriched PMDs. In addition, the mean methylation level of H3K9me3-enriched PMDs was slightly but significantly lower than that of H3K27me3 enriched PMDs (*S1F Fig*, p = 6.85x10$^{-13}$, Wilcoxon rank sum test) and the mean replication timing of H3K9me3-enriched PMDs was significantly later than H3K27me3-enriched PMDs (*S1G Fig*, p < 2.2x10$^{-16}$, Wilcoxon rank sum test).

We then asked how the methylome changed in DNMT1 KO cells. Global mean methylation levels measured by WGBS in DNMT1 KO cells were 59.2% as compared to 75.7% in HCT116 cells (*S1H Fig*), a similar reduction to previous reports [26]. Despite this global reduction, mean methylation levels across the genome in DNMT1 KO cells had a bimodal distribution like that observed in HCT116 cells (Fig 1B-1C).

To understand whether different genomic domains might display differential requirement for DNMT1, we then compared the degree of methylation loss observed in DNMT1 KO cells between HCT116 PMDs and HMDs. While both PMDs and HMDs had reduced methylation in DNMT1 KO cells, the level of methylation loss in PMDs observed in DNMT1 KO cells was significantly greater than in HMDs (Fig 1D, p < 2.2x10$^{-16}$, Wilcoxon rank sum test).

To understand how these differences in the degree of methylation loss related to the properties of PMDs, we then analysed how mean methylation changes across the genome related to replication timing and PMD-associated histone marks in HCT116 cells. Despite the general late replication of PMDs, we observed a weak correlation between DNA methylation reduction and HCT116 replication timing measured by repli-seq (Fig 1E, Rho = 0.048, p < 2.2x10$^{-16}$, Spearman's rank correlation). In contrast we observed stronger, significant correlations between mean H3K9me3 or H3K27me3 levels and DNA methylation reduction in DNMT1 KO cells (Fig 1F-1G, Rho = -0.136 and Rho = -0.428 respectively, both p < 2.2x10$^{-16}$, Spearman's rank correlations).

Taken together, these analyses suggest that the loss of methylation observed upon removal of DNMT1 in cancer cells is non-uniform and DNA methylation is preferentially lost from PMDs associated with the histone marks H3K27me3 and H3K9me3.

## A subset of H3K9me3-marked PMDs are hypermethylated in DNMT1 knockout cells

When examining differences in the methylation levels of PMDs and HMDs between HCT116 and DNMT1 KO cells, we noted that a subset of PMDs had higher levels of DNA methylation in DNMT1 KO cells (Fig 2A-2B). This was surprising given the reduced function of the major maintenance DNMT in these cells and the bias towards greater methylation losses in PMDs in DNMT1 KO cells.

To understand the factors that led to this increase in methylation at some PMDs upon removal of DNMT1, we defined hypermethylated PMDs as those where mean methylation levels in DNMT1 KO cells were ≥ 5% higher than in HCT116 cells (60 out of 546 PMDs, 10.9%) (Figs 2B *and S2A*). Gains of methylation were significantly more likely to occur at PMDs than HMDs and only 3 out of 556 HMDs had ≥ 5% mean methylation in DNMT1 KO cells compared to HCT116 cells (p = 6.33x10$^{-16}$, Fisher's exact test). These hypermethylated PMDs were distributed across all the autosomal chromosomes and were longer than all other HCT116 PMDs (*S2B Fig*, median length 2.12Mb versus 0.94Mb of all other PMDs, p = 1.52x10$^{-5}$, Wilcoxon rank sum test).

We then asked whether hypermethylated PMDs were associated with a distinct set of chromatin marks compared to all other HCT116 PMDs. Hypermethylated PMDs had similar levels of H3K9me3 but lower levels of H3K27me3 than all other HCT116 PMDs (Figs 2B-2C *and S2C-S2D*, p = 0.59 and <2.2x10$^{-16}$ respectively, Wilcoxon rank sum tests). Hypermethylated PMDs also replicated slightly but significantly earlier than all other HCT116 PMDs (Figs 2B-2C *and S2E*). Consistent with these characteristics, we found that the hypermethylated PMDs were significantly enriched in H3K9me3-marked PMDs defined in our previous analysis (43 out of 60 hypermethylated PMDs were H3K9me3-marked PMDs, 71.66%, p = 2.42x10$^{-5}$, Fisher's exact test).

These observations suggest a subset of H3K9me3-marked PMDs have higher levels of methylation in DNMT1 KO cells despite their severe reduction of maintenance methyltransferase activity.

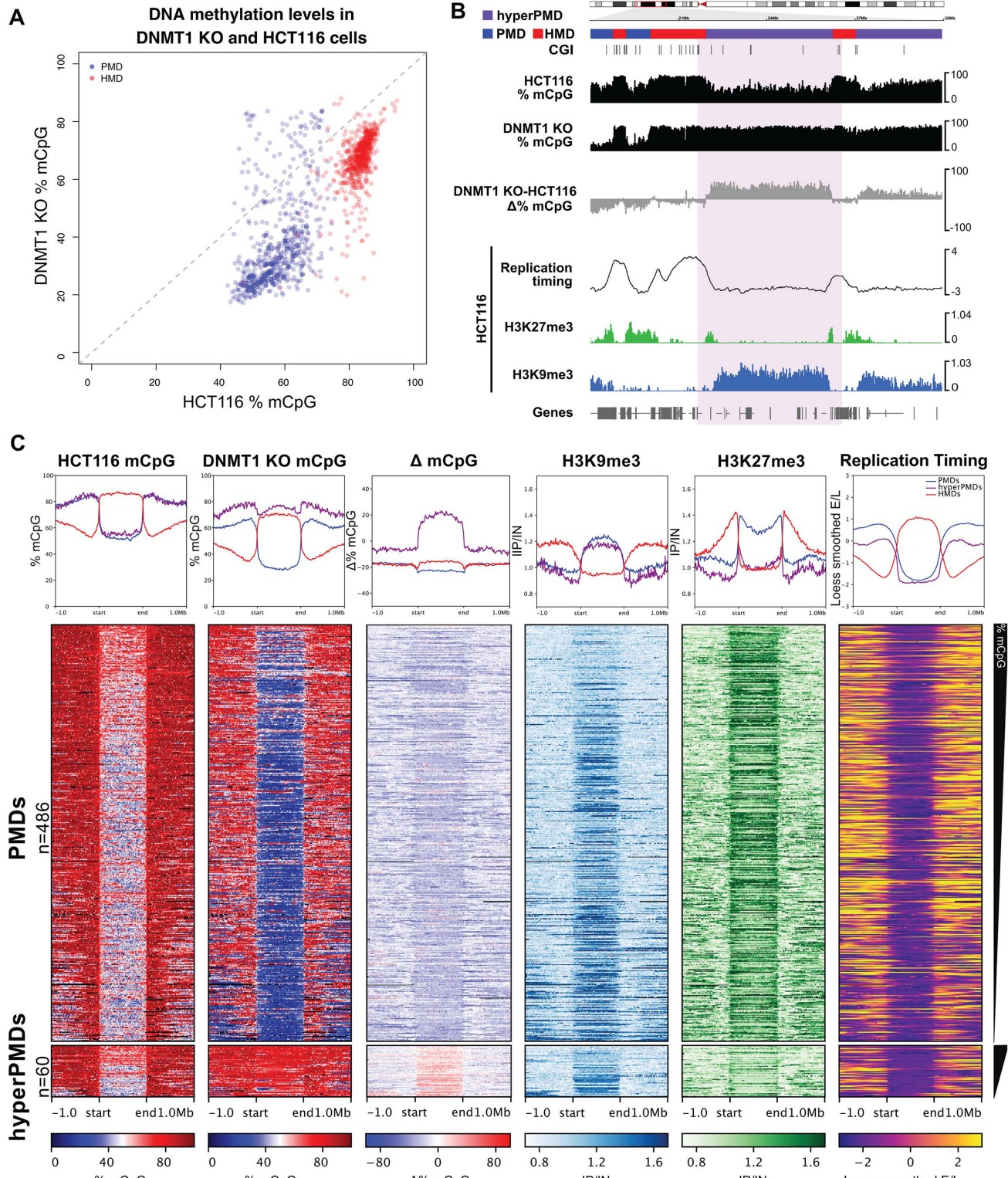

**Fig 2. A subset of H3K9me3-marked PMDs are hypermethylated in DNMT1 knockout cells.** (**A**) Scatter plot of mean methylation levels at PMDs and HMDs in DNMT1 KO cells versus HCT116 cells. (**B**) Representative genomic loci showing changes of DNA methylation at a hypermethylated PMD in DNMT1 KO cells. Genome browser plots showing absolute (black) and differential (grey) DNA methylation levels (mCpG) alongside HCT116 histone

modification ChIP-seq and repli-seq. DNA methylation levels are plotted in 10 kb genomic windows. ChIP-seq tracks are normalised $\log_{10}$ IP/IN. Replication timing data are loess smoothed repli-seq early/late ratios over 10 kb. Representative hypermethylated PMD and HMD are indicated by the coloured boxes. CGI = CpG islands. Chromosomal co-ordinates shown are from hg38: chr9:18,000,000-30,000,000. **(C)** Heatmaps and pileup plots of HCT116 H3K9me3, H3K27me3 and DNA methylation levels alongside replication timing for hypermethylated PMDs (n = 60 domains) and all other HCT116 PMDs (n = 486 domains). ChIP-seq data are mean normalised IP/IN. DNA methylation levels are mean % mCpG. Replication timing data are mean loess smoothed repli-seq early/late ratios over 10kb. PMDs are aligned and scaled to the start and end points of each domain and ranked based on their mean methylation levels in HCT116 cells. All histone ChIP-seq and repli-seq data shown are derived from the mean of two biological replicates.

## The replication timing of hypermethylated PMDs remains similar in DNMT1 KO cells

We next sought to understand the factors that could have led to the higher levels of methylation observed at hypermethylated PMDs in DNMT1 KO cells. The late replication of PMDs in S-phase is proposed to underpin their loss of methylation with successive cell divisions [17,24,30]. We therefore wondered whether the higher methylation levels of hypermethylated PMDs was linked to a shift in their replication timing.

We compared the replication programs of DNMT1 KO and HCT116 cells using repli-seq (Fig 3A). The overall replication timing of DNMT1 KO cells was highly significantly correlated to that of HCT116 cells (S3A Fig, Rho = 0.91, p < 2.2x10$^{-16}$, Spearman's rank correlation) suggesting replication programs were not grossly altered. As in HCT116 cells, DNMT1 KO cell replication timing was significantly but weakly correlated with methylation levels consistent with highly methylated parts of the DNMT1 KO genome having earlier replication timing than lowly methylated regions (S3B-S3C Fig, HCT116: Rho = 0.47, DNMT1 KO: Rho = 0.27, both p < 2.2x10$^{-16}$, Spearman's rank correlations). Similar to our observations in HCT116 cells, HCT116 PMDs were significantly later replicating than HCT116 HMDs in DNMT1 KO cells (S1A and S3D Figs, p < 2.2x10$^{-16}$, Wilcoxon rank sum test).

We then focused specifically on understanding whether the replication timing of hypermethylated PMDs changed in DNMT1 KO cells compared to other HCT116 H3K9me3-marked PMDs. Despite their high levels of methylation in DNMT1 KO cells, overall, hypermethylated PMDs remained late replicating in DNMT1 KO cells (Fig 3A-3C). In fact, hypermethylated PMDs were observed to replicate slightly but significantly later in DNMT1 KO cells compared to HCT116 cells (Fig 3B, p < 2.2x10$^{-16}$, Wilcoxon rank sum test). In contrast, other H3K9me3-enriched HCT116 PMDs replicated slightly but significantly earlier in DNMT1 KO cells than HCT116 cells (Fig 3B, p < 2.2x10$^{-16}$, Wilcoxon rank sum test).

Overall, these analyses suggest that despite the global associations between late replication timing and low levels of methylation, hypermethylated PMDs do not become earlier replicating in DNMT1 KO cells.

## DNMT3A localises to hypermethylated PMDs

We next asked whether the higher methylation levels at hyper PMDs in DNMT1 KO cells might result from activity of the *de novo* methyltransferases, DNMT3A or DNMT3B.

To understand whether this might be the case, we first examined whether DNMT3A or DNMT3B were differentially expressed between HCT116 and DNMT1 KO cells using RNA-seq data. Consistent with the knockout of DNMT1, DNMT1 RNA levels were significantly reduced in DNMT1 KO cells compared to HCT116 cells (S4A Fig, logFC = -2.88, p = 8.42x10$^{-9}$, Benjami-Hochberg adjusted F-test). However, only small differences in the RNA levels of both DNMT3A and DNMT3B were observed in DNMT1 KO cells relative to HCT116 cells (S4A Fig). We then asked whether this was reflected at the protein level using Western blots, finding that DNMT3A was expressed to similar levels in HCT116 and DNMT1 KO cells (S4B Fig). In contrast, catalytically active DNMT3B2 was downregulated in DNMT1 KO cells but its catalytically inactive adaptor isoform, DNMT3B3, remained similarly expressed (S4C Fig). This suggests DNMT3A levels are not greatly altered in DNMT1 KO cells, but that DNMT3B2 is downregulated.

We then asked whether DNMT3A or DNMT3B localised differently in HCT116 and DNMT1 KO cells. DNMT3A and DNMT3B antibodies are unsuitable for ChIP. However, we have previously shown that ectopically expressed DNMT3B

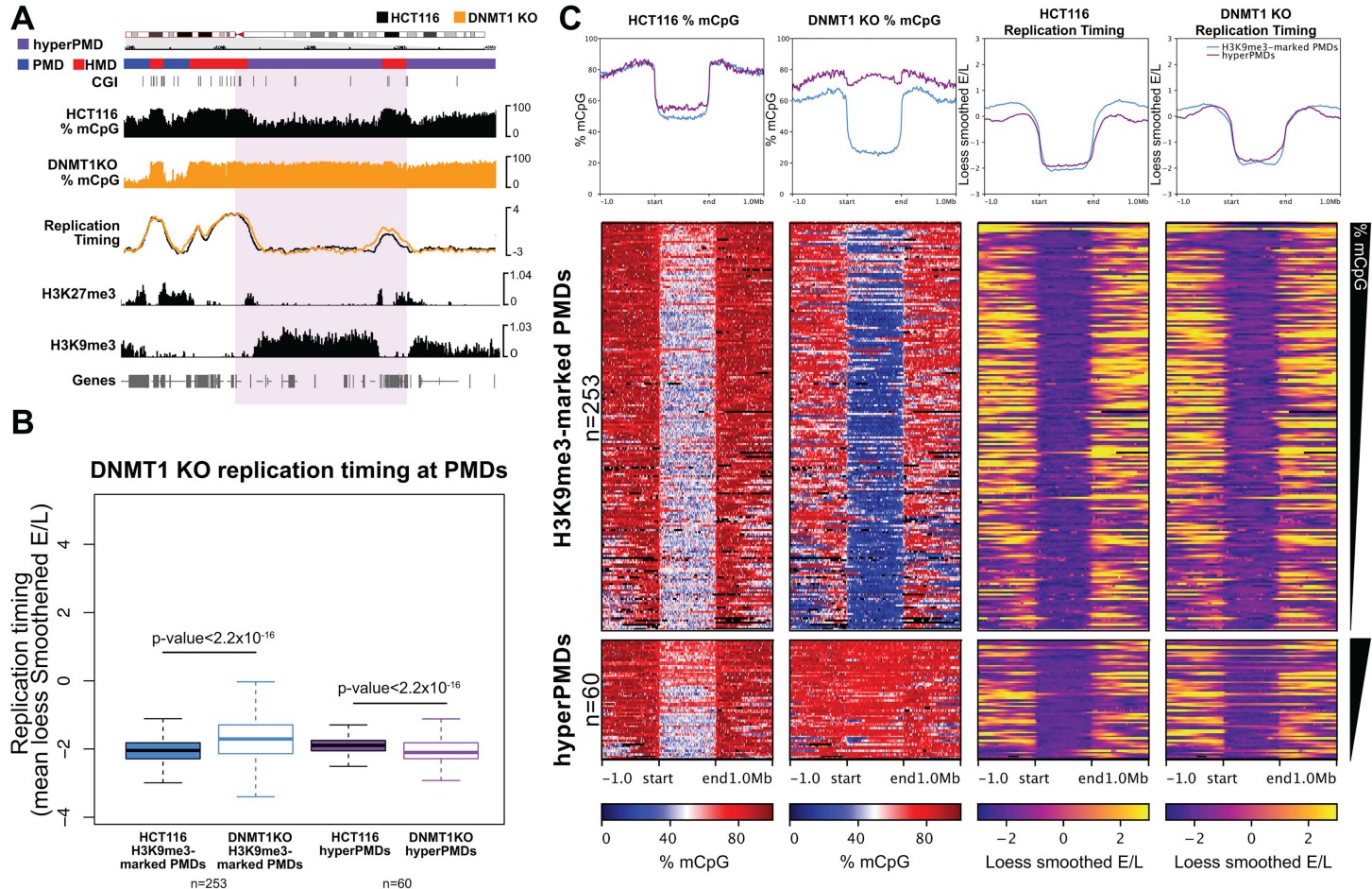

**Fig 3. The replication timing of hypermethylated PMDs remains similar in DNMT1 KO cells. (A)** Representative genomic loci showing DNA methylation and replication timing HCT116 and DNMT1 KO cells. Genome browser plots showing DNA methylation levels (mCpG) alongside HCT116 histone modification ChIP-seq and repli-seq. DNA methylation levels are plotted in 10 kb genomic windows. ChIP-seq tracks are normalised $\log_{10}$ IP/IN. Replication timing data are loess smoothed repli-seq early/late ratios over 10 kb. Representative hypermethylated PMD is indicated by the coloured box. CGI = CpG islands. Chromosomal co-ordinates shown are from hg38: chr9:18,000,000-30,000,000. **(B)** Boxplot comparing HCT116 and DNMT1 KO replication timing at hypermethylated PMDs (n = 60 domains) and other H3K9me3-marked PMDs (n = 253 domains). Replication timing data are mean loess smoothed repli-seq early/late ratios over 10kb. Box plot annotation: lines = median; box = 25th–75th percentile; whiskers = 1.5 × interquartile range from box. P-values are from two-sided Wilcoxon rank sum tests. **(C)** Heatmaps and pileup plots of HCT116 and DNMT1 KO DNA methylation levels alongside replication timing for hypermethylated PMDs (n = 60 domains) and all other H3K9me3-marked PMDs (n = 253 domains). ChIP-seq data are mean normalised IP/IN. DNA methylation levels are mean % mCpG. Replication timing data are mean loess smoothed repli-seq early/late ratios over 10kb. PMDs are aligned and scaled to the start and end points of each domain and ranked based on their mean methylation levels in HCT116 cells. All histone ChIP-seq and repli-seq data shown are derived from the mean of two biological replicates.

recapitulates the localisation of endogenous DNMT3B in HCT116 cells [31] suggesting that tagged, ectopically expressed enzymes can be used to understand DNMT3A and DNMT3B recruitment. Therefore, we expressed T7-tagged DNMT3A or DNMT3B in HCT116 and DNMT1 KO cells and conducted ChIP-seq to determine their localisation. For this experiment we selected the main catalytically active versions of the enzymes found in HCT116 cells, DNMT3A1 and DNMT3B2 [31,32].

In HCT116 cells, both DNMT3A and DNMT3B levels were significantly higher in HMDs compared to PMDs (*S4D*-*S4F Fig*). HCT116 DNMT3B ChIP-seq signal was significantly correlated with H3K36me3 ChIP-seq signal (*S4G Fig*, Rho = 0.54, p < 2.2x10⁻¹⁶, Spearman's rank correlation) whereas HCT116 DNMT3A ChIP-seq signal showed lower

correlations with H3K36me3 (*S4H Fig*, Rho = 0.12, p < 2.2x10$^{-16}$, Spearman's rank correlation), as previously reported [33,34].

We next asked how DNMT3A and DNMT3B localised in DNMT1 KO cells as compared to HCT116 cells. The overall profiles of DNMT3A and DNMT3B in DNMT1 KO cells were equivalent to that observed in HCT116 cells with significantly more of both enzymes observed in HMDs compared to PMDs (*S4D-S4F Fig*). In addition, both DNMT3A and DNMT3B ChIP-seq signals across the genome were significantly correlated between HCT116 and DNMT1 KO cells (*S4I-S4J Fig*, both Rho = 0.64, p < 2.2x10$^{-16}$, Spearman's rank correlations).

To understand how DNMT3A and DNMT3B might contribute to the higher levels of methylation at hypermethylated PMDs in DNMT1 KO cells, we examined their localisation to these domains. In HCT116 cells, levels of both enzymes in hypermethylated PMDs were low and equivalent to other HCT116 H3K9me3-marked PMDs (*Fig 4A-4D*). Both enzymes were depleted from HCT116 H3K9me3-marked PMDs in DNMT1 KO cells, but DNMT3A levels were significantly increased at hypermethylated PMDs in DNMT1 KO cells when compared to HCT116 cells (*Fig 4A-4B, 4D*, p < 2.2x10$^{-16}$, Wilcoxon rank sum test). Despite this increase, DNMT3A levels at hypermethylated PMDs in DNMT1 KO cells were lower than was observed in the surrounding HMD regions in DNMT1 KO cells (*Figs 4A-4B and S4D-S4E*). DNMT3B levels were not significantly increased at hypermethylated PMDs in DNMT1 KO cells (*Fig 4A, 4C, 4D*, p = 0.83, Wilcoxon rank sum test) but DNMT3B levels at other H3K9me3-marked PMDs were significantly reduced in DNMT1 KO cells compared to HCT116 cells (*Fig 4C, p < 2.2x10$^{-16}$*).

To further understand how *de novo* DNMT activity is distributed in the cancer genome, we analysed DNA methylation levels following acute DNMT1 removal from a DNMT1 degron HCT116 line, AID-DNMT1 cells [35]. DNMT1 removal leads to genome-wide hypomethylation [35]. In the absence of *de novo* activity this loss would be expected to follow exponential decay dynamics irrespective of initial level (*S5A Fig*). However, differential *de novo* DNMT activity between genomic domains would alter their relative kinetics (*S5A Fig*). Specifically, higher levels of *de novo* DNMT activity i) slows DNA methylation loss and ii) results in DNA methylation levels plateauing at a higher level (*S5A Fig*).

We analysed a time-course of WGBS following DNMT1 removal from AID-DNMT1 cells, observing that both HMDs and PMDs lost methylation and that levels plateaued after 6 days. However, HMDs lost methylation at a significantly slower rate than PMDs and remained at a significantly higher DNA methylation level following completion of the time-course (*S5B-S5C Fig*, both p < 2.2x10$^{-16}$, Wilcoxon rank sum tests) suggesting higher *de novo* DNMT activity in HMDs than PMDs. To understand whether *de novo* DNMT activity in HCT116 cells differed at hypermethylated PMDs relative to other PMDs, we also analysed their kinetics. Relative to other H3K9me3-marked PMDs, the observed rate of DNA methylation loss was significantly slower at hypermethylated PMDs and they retained significantly higher level of DNA methylation (*S5D Fig*, p = 0.027 and 2.77x10$^{-4}$ respectively, Wilcoxon rank sum tests). This is consistent with hypermethylated PMDs having higher *de novo* DNMT activity than other PMDs in HCT116 cells.

This suggests that the higher levels of DNA methylation at hypermethylated PMDs in DNMT1 KO cells are associated with DNMT3A localisation.

## Hypermethylated PMDs lose H3K9me3 and gain H3K36me2

We next sought to understand why DNMT3A was recruited to hypermethylated PMDs in DNMT1 KO cells. The recruitment of DNMTs to the genome is largely driven by the recognition of histone marks [36]. However, DNMT3A has also previously been reported to be excluded from heterochromatin [37,38].

We therefore conducted ChIP-seq for the heterochromatic histone marks H3K9me3 and H3K27me3 in DNMT1 KO cells and compared their profile to HCT116 cells. The overall profiles of H3K9me3 and H3K27me3 were significantly correlated between the two cell types (*S6A-S6C Fig*, Rho = 0.56 and 0.71 respectively, both p < 2.2x10$^{-16}$, Spearman's rank correlations). In support of a broadly similar genome-wide distribution of the two marks, Hidden Markov Model defined H3K9me3 and H3K27me3 domains in the two cell types also significantly overlapped (H3K9me3: Jaccard = 0.44, Fisher's

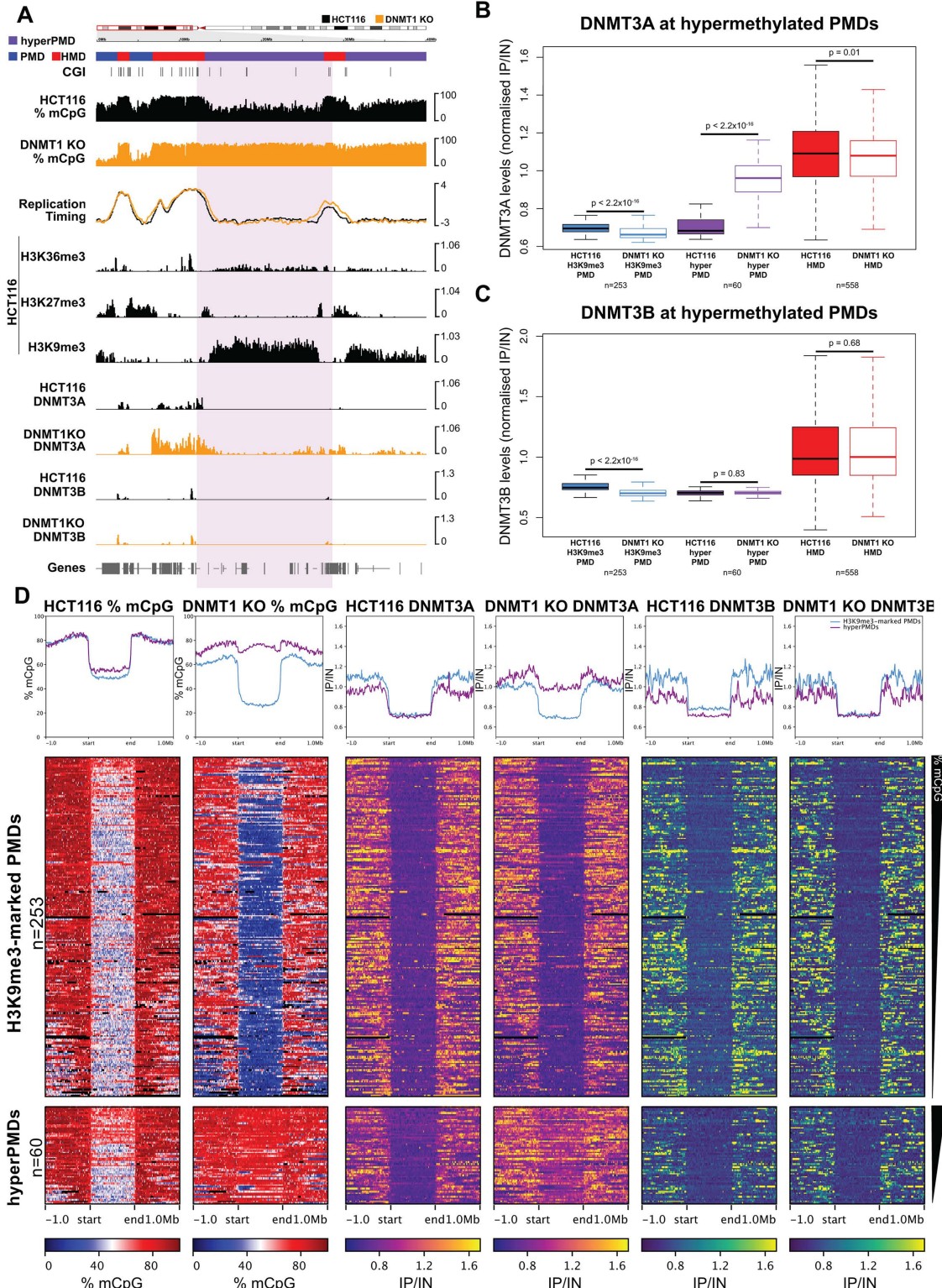

**Fig 4. DNMT3A localises to hypermethylated PMDs. (A)** Representative genomic loci showing DNMT3A localisation at a hypermethylated PMD in DNMT1 KO cells. Genome browser plots showing DNA methylation levels (mCpG) alongside DNMT3A/B ChIP-seq and HCT116 histone modifications and repli-seq. DNA methylation levels are plotted in 10 kb genomic windows. ChIP-seq tracks are normalised $\log_{10}$ IP/IN. Replication timing data are

loess smoothed repli-seq early/late ratios over 10 kb. Representative hypermethylated PMD is indicated by the coloured box. CGI = CpG islands. Chromosomal co-ordinates shown are from hg38: chr9:18,000,000-30,000,000. **(B, C)** Boxplots showing DNMT3A (**B**) and DNMT3B (**C**) levels at different domains in HCT116 and DNMT1 KO cells. Shown are hypermethylated PMDs (n = 60 domains) compared to other H3K9me3-marked PMDs (n = 253 domains) and HMDs (n = 558 domains). ChIP-seq data are mean normalised IP/IN. **(D)** Heatmaps and pileup plots of HCT116 and DNMT1 KO DNA methylation levels alongside DNMT3A/B for hypermethylated PMDs (n = 60 domains) and all other H3K9me3-marked PMDs (n = 253 domains). ChIP-seq data are mean normalised IP/IN. DNA methylation levels are mean % mCpG. PMDs are aligned and scaled to the start and end points of each domain and ranked based on their mean methylation levels in HCT116 cells. For boxplots: Lines = median; box = 25th–75th percentile; whiskers = 1.5 × interquartile range from box. All p-values are from two-sided Wilcoxon rank sum tests. All histone ChIP-seq and repli-seq data shown are derived from the mean of two biological replicates.

test p < 2.2x10^-16, H3K27me3 Jaccard = 0.57, Fisher's test p < 2.2x10^-16). However, we noted that levels of H3K9me3 in H3K9me3-marked PMDs were slightly reduced in DNMT1 KO cells and occasional spreading of the H3K9me3 mark was observed (Figs 5A and S6A).

We then went on to specifically ask whether H3K9me3 or H3K27me3 were altered at hypermethylated PMDs in DNMT1 KO cells. We observed that hypermethylated PMDs showed a significant decrease in both H3K9me3 and H3K27me3 levels in DNMT1 KO cells compared to HCT116 cells (Figs 5A-5D and S6D, p = 2.06x10^-6, Wilcoxon rank sum test). We also observed a decrease in H3K9me3 levels from H3K9me3-marked PMDs in DNMT1 KO cells (Figs 5A, 5C and S6A). However, levels of H3K9me3 in hypermethylated PMDs decreased to that of background while levels at other H3K9me3-marked PMDs remained above background (Figs 5A-5B and S6A). This was supported by the observation that hypermethylated PMDs were significantly less likely to overlap H3K9me3 domains in DNMT1 KO cells than other H3K9me3-marked PMDs (11 of 60, 18.33%, hypermethylated PMDs overlap DNMT1 KO H3K9me3 domains whereas 192 of 210, 91.42%, of HCT116 H3K9me3-marked PMDs do, Fisher's test p < 2.2x10^-16). Hypermethylated PMDs also showed a significant decrease in H3K27me3 levels in DNMT1 KO cells as compared to HCT116 cells (Figs 5B, 5D, and S6D, p = 1.71x10^-10, Wilcoxon rank sum test). This reflected the loss of H3K27me3 marking the borders of hypermethylated PMDs in HCT116 (S6D Fig). Overall, this suggests that hypermethylated PMDs lose heterochromatic histone marks in DNMT1 KO cells.

DNMT3A also localises in the genome through the association of its PWWP domain with H3K36me2 (Weinberg et al., 2019). To understand whether differences in H3K36me2 might be involved in the localisation of DNMT3A to hypermethylated PMDs, we therefore performed ChIP-seq for H3K36me2 in HCT116 and DNMT1 KO cells. The distribution of H3K36me2 in DNMT1 KO cells was significantly correlated with that in HCT116 cells (S7A Fig, Rho = 0.68, p < 2.2x10^-16, Spearman's rank correlation) and in both HCT116 and DNMT1 KO cells we observed that H3K36me2 was primarily associated with euchromatic HMD regions and depleted from PMDs (S6A Fig). Consistent with this, H3K36me2 levels were also significantly negatively correlated with the PMD-associated marks H3K27me3 and H3K9me3 in both HCT116 and DNMT1 KO cells (S7B-S7C Fig).

We then specifically examined H3K36me2 levels at hypermethylated PMDs. Like other H3K9me3-marked PMDs, hypermethylated-PMDs had low levels of H3K36me2 in HCT116 cells (Fig 5A-5B). In DNMT1 KO cells, H3K36me2 levels remained low in H3K9me3-marked PMDs (Fig 5A-5B). However, hypermethylated PMDs specifically showed significant increases of H3K36me2 in DNMT1 KO cells compared to HCT116 cells (Fig 5A-5B, and 5E, p < 2.2x10^-16, Wilcoxon sum test).

To understand whether increases in the levels of an H3K36me2 methyltransferase might be responsible for increases in this mark at hypermethylated PMDs in DNMT1 KO cells, we examined RNA levels of putative H3K36 methyltransferases in our RNA-seq data. Robust evidence exists for 5 different human H3K36 methyltransferases: NSD1, NSD2, NSD3, ASH1L, SETD2 [39], and their deletion in human cells entirely removes H3K36 methylation [40]. All 5 were expressed in both HCT116 and DNMT1 KO cells but their RNA levels were not significantly different between the two cell lines (S7D Fig) meaning that it is unclear which might be responsible for deposition of H3K36me2 at hypermethylated PMDs.

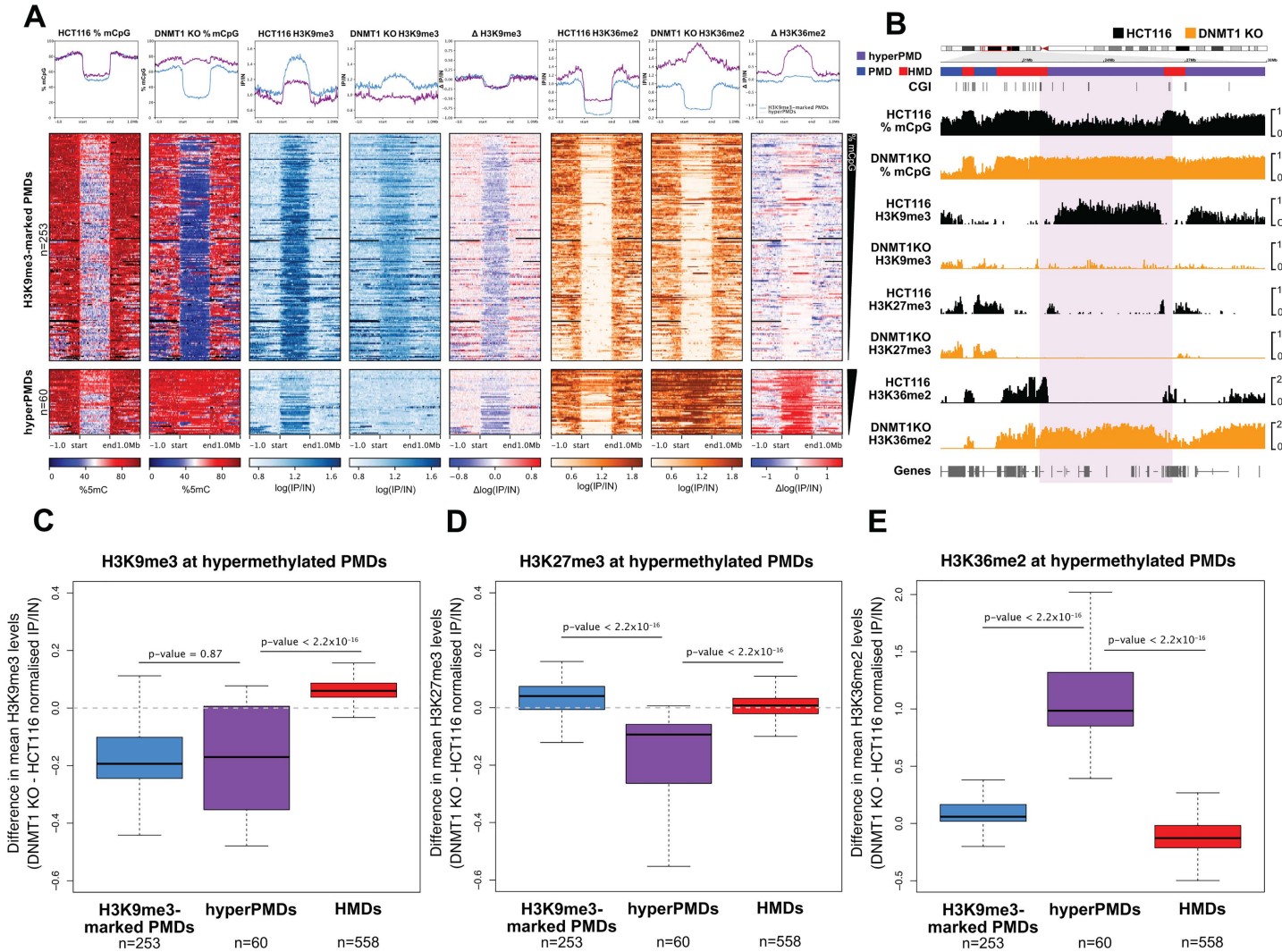

**Fig 5. Hypermethylated PMDs lose H3K9me3 and gain H3K36me2. (A)** Heatmaps and pileup plots of HCT116 and DNMT1 KO DNA methylation levels alongside H3K9me3 and H3K36me2 levels for hypermethylated PMDs (n = 60 domains) and all other H3K9me3-marked PMDs (n = 253 domains). ChIP-seq data are mean normalised IP/IN. DNA methylation levels are mean % mCpG. PMDs are aligned and scaled to the start and end points of each domain and ranked based on their mean methylation levels in HCT116 cells. **(B)** Representative genomic loci showing H3K9me3 and H3K36me2 levels at a hypermethylated PMD in DNMT1 KO cells. Genome browser plots showing DNA methylation levels (mCpG) alongside H3K9me3 and H3K36me2 ChIP-seq. DNA methylation levels are plotted in 10 kb genomic windows. ChIP-seq tracks are normalised $\log_{10}$ IP/IN. Representative hypermethylated PMD is indicated by the coloured box. CGI = CpG islands. Chromosomal co-ordinates shown are from hg38: chr9:18,000,000-30,000,000. **(C-E)** Boxplots showing change in histone modification levels in DNMT1 KO cells. **(C)** Change in H3K9me3 levels. **(D)** Change in H3K27me3 levels. **(E)** Change in H3K36me2 levels. Plots compare H3K9me3-marked PMDs (n = 253 domains), hypermethylated PMDs (n = 60 domains) and HMDs (n = 558 domains). ChIP-seq data are difference in mean normalised IP/IN between DNMT1 KO and HCT116 cells. For boxplots: Lines = median; box = 25th–75th percentile; whiskers = 1.5 × interquartile range from box. All p-values are from two-sided Wilcoxon rank sum tests. All histone ChIP-seq data shown are derived from the mean of two biological replicates.

We then asked whether this remodelling of chromatin structure was permanent by expressing human DNMT1 in DNMT1 KO cells and assaying genome-wide DNA methylation levels using Nanopore sequencing. DNMT1 expression led to a mean 4.83% increase in DNA methylation levels that was not observed in control cells expressing eGFP (where the mean increase was 0.22%, *S8A Fig*). However, methylation levels did not return to those of HCT116 cells (*S8A Fig*)

consistent with the predominant role of DNMT1 being a maintenance rather than a *de novo* methyltransferase [36]. These gains of methylation were observed throughout the genome with a bias towards slightly greater increases in HMDs versus PMDs (*S8B*-*S8C Fig*, p = 1.85x10$^{-5}$, Wilcoxon rank sum test). However, we did not observe a reversion of methylation levels at hypermethylated PMDs which gained slightly more methylation than other H3K9me3-marked PMDs (*S8D Fig*, p = 0.04, Wilcoxon rank sum test). This suggests that the generation of DNMT1 KO cells led to a reconfiguration of the chromatin landscape at hypermethylated PMDs that does not revert upon reintroduction of DNMT1. Furthermore, as Nanopore sequencing can distinguish methylcytosine from hydroxymethylcytosine [41], unlike WGBS [42], these data confirm that the changes we report are due to differences in the abundance of methylcytosine.

Taken together, this suggests that a subset of H3K9me3-marked PMDs from HCT116 cells reconfigure their chromatin state, losing this mark in DNMT1 KO cells in association with gain of H3K36me2, leading to the recruitment of DNMT3A and subsequent higher levels of DNA methylation.

## Genes in hypermethylated PMDs are upregulated

We then sought to understand whether changes in chromatin structure at hypermethylated PMDs might impact on gene expression. To do so we examined gene expression levels for genes wholly located within different types of genomic domains using RNA-seq from HCT116 and DNMT1 KO cells.

As previously reported [1,10], we observed that genes within PMDs had significantly lower expression levels than those located in HMDs in HCT116 cells (*S9A Fig*, p = 7.96x10$^{-11}$, Wilcoxon rank sum test). This remained the case in DNMT1 KO cells (*S9A Fig*, p = 5.89x10$^{-10}$, Wilcoxon rank sum test).

We then asked whether loss of DNMT1 and the resulting remodelling of DNA methylation led to changes in the expression of individual genes. Significantly upregulated genes between DNMT1 KO and HCT116 cells were significantly enriched in HCT116 PMDs compared to HMDs (*S9B Fig*, p = 0.0023, Fishers exact test) whereas significantly downregulated genes were not (*S9B Fig*, p = 0.24, Fishers exact test). However, we found that the majority (95.39%) of PMD genes were filtered out of the analysis in standard RNA-seq pipelines due to their low expression value across all samples. This left only 57 genes located entirely within PMDs that could be examined for differential regulation and precluded assessment of genes located in hypermethylated PMDs as only 1 gene located within a hypermethylated PMD remained in the analysis.

To more robustly analyse the overall behaviour of PMD-located genes, we therefore, re-calculated log fold-changes for all genes without implementing a filtering step to remove genes with low expression across all samples. This increased the number of PMD genes analysed to 1,235. Similarly to our analysis of differential expression at individual genes (*S9B Fig*), we observed that genes located within PMDs were upregulated to a significantly greater level in DNMT1 KO cells relative to those located within HMDs (*S9C Fig*, p = 1.24x10$^{-9}$, Wilcoxon rank sum test). Furthermore, we observed that the 14 genes located wholly within hypermethylated PMDs were upregulated to a significantly greater level than the 596 genes located within other HCT116 H3K9me3-marked PMDs (*S9D Fig*, p = 0.0014, Wilcoxon rank sum test).

Overall, this suggests that genes located in hypermethylated PMDs are differentially regulated compared to genes in other PMDs.

## Hypermethylated PMDs have variable methylation levels in colorectal tumours

To understand what our findings tell us about methylome organisation in clinical colorectal tumours, we collected published whole genome methylome sequencing from 45 colorectal tumours and 10 normal colon samples [9,43–47] and examined mean DNA methylation levels in HCT116 PMDs compared to HMDs. This revealed that HCT116 PMDs had significantly lower mean DNA methylation levels in colorectal tumours than HCT116 HMDs (Fig 6A-6B, p = 5.68x10$^{-14}$, Wilcoxon rank sum test). In addition, HCT116 PMDs had lower mean DNA methylation levels in colorectal tumours than in the normal colon (Fig 6A-6B). This suggests that domain-level methylation patterns observed in HCT116 cells are

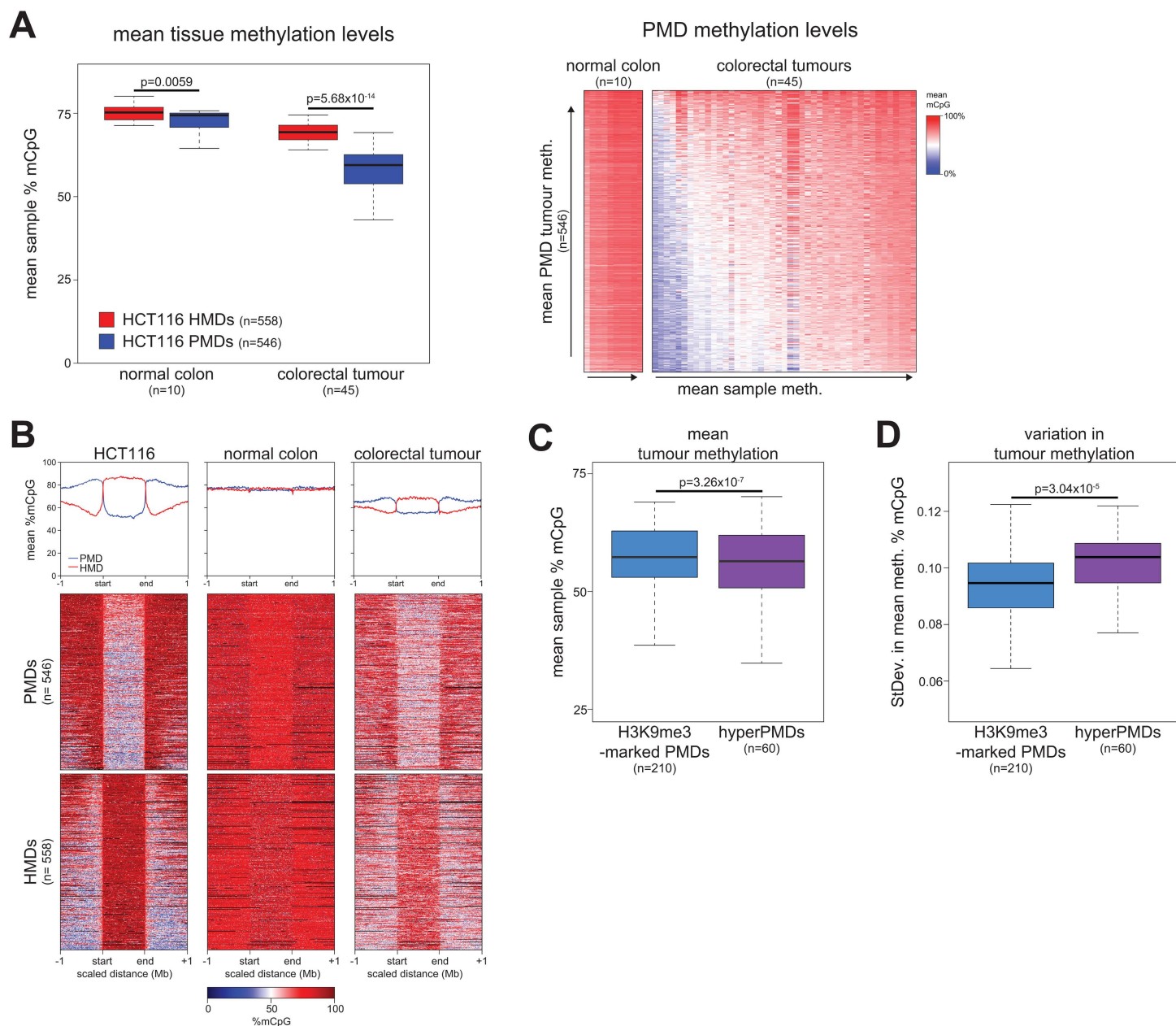

**Fig 6. Hypermethylated PMDs have variable methylation levels in colorectal tumours. (A)** HCT116 PMDs are hypomethylated in colorectal tumours. Left, boxplot of mean levels of PMD and HMD methylation for normal colon and colorectal tumour samples (n = 10 and 45 samples respectively). Right, heatmap of mean DNA methylation levels at individual PMDs (n = 546 domains) in normal colon and colorectal tumour samples. PMDs are ordered by their mean methylation in colorectal tumours and samples are ordered by their mean methylation in all HCT116 PMDs. **(B)** Representative heatmaps and pileup plots of HCT116, normal colon and a colorectal tumour DNA methylation levels for HCT116 PMDs (n = 546 domains) and HMDs (n = 558 domains). DNA methylation levels are mean % mCpG. PMDs and HMDs are aligned and scaled to the start and end points of each domain and ranked based on their mean methylation levels in HCT116 cells. Example normal colon and tumour data re-analysed from a previous publication [9]. **(C,D)** Boxplots of mean methylation (**C**) and standard deviation in methylation level (**D**) across colorectal tumours (n = 45) for H3K9me3-marked PMDs (n = 210 domains) and hypermethylated PMDs (n = 60 domains). For boxplots, lines = median; box = 25th–75th percentile; whiskers = 1.5 × interquartile range from box. All p-values are from two-sided Wilcoxon rank sum tests.

consistent with those in clinical colorectal tumours. In the normal colon HCT116 PMDs were slightly but significantly hypomethylated relative to HCT116 HMDs (Fig 6A, p = 0.0059, Wilcoxon rank sum test) potentially consistent with age-associated PMD hypomethylation [24].

We then assessed whether any differences in methylation was observable between HCT116 H3K9me3-marked PMDs and hypermethylated PMDs. Hypermethylated PMDs had slightly but significantly lower mean methylation levels than H3K9me3-marked PMDs (Fig 6C, p = 3.26x10$^{-7}$, Wilcoxon rank sum test). However, we found that mean methylation level at hypermethylated PMDs was significantly more variable across colorectal tumours than for H3K9me3-marked PMDs (Fig 6D, p = 3.04x10$^{-5}$, Wilcoxon rank sum test). This observation is consistent with the hypothesis that chromatin structure and thus DNA methylation levels is more labile at hypermethylated PMDs *in vivo* in clinical tumours.

Taken together this suggests that the large-scale methylome organisation of HCT116 cells reflects that of colorectal tumours and that hypermethylated PMDs may be more labile than other PMDs *in vivo*.

## Discussion

Here we have analysed the effect of DNMT1 knockout on PMDs in colorectal cancer cells. We find that losses of methylation in DNMT1 KO cells are biased towards PMDs. However, we also observe that several H3K9me3-marked regions have increased levels of methylation in these cells and lose PMD identity. In association with this, these hypermethylated PMDs lose H3K9me3, gain H3K36me2 and recruit the *de novo* methyltransferase DNMT3A (Fig 7A). These observations suggest a key role for differential *de novo* methyltransferase activity in determining which parts of the genome become PMDs in cancer. We propose that the activity of DNMT1 is not sufficient to maintain DNA methylation levels throughout the genome and that re-iterative activity of DNMT3A and DNMT3B prevents euchromatic regions becoming hypomethylated. In contrast, a lack of this *de novo* activity leads to the formation of PMDs in heterochromatic regions in tumours (Fig 7B).

Previous work has connected the hypomethylation of PMDs to their late replication in S-phase [24]. Analysis of newly replicated DNA by bisulfite sequencing suggests that re-methylation can take several hours [20] and may occur in two phases [21]. This is supported by mass-spectrometry analysis suggesting that some re-methylation occurs post-mitosis [22]. These studies have led to the hypothesis that there is insufficient time to maintain DNA methylation patterns in late replicating regions in rapidly dividing cancer cells resulting in their hypomethylation. Here we do observe that DNMT1 knockout in a colorectal cancer cell line biases DNA methylation losses to PMDs which are late replicating. This parallels a recent study reporting that losses of DNA methylation upon shRNA depletion of DNMT1 in HCT116 cells are biased to low-CpG density regions that are enriched in PMDs [48]. However, we observe weak correlations between DNA methylation loss and replication timing in DNMT1 KO cells. Furthermore, we observe that some regions of the genome gain DNA methylation in these cells while remaining late replicating. Overall, our results suggest that the formation of PMDs is not tightly linked to the replication timing program.

We instead observe stronger correlations between methylation levels and localisation of the *de novo* DNMTs. Normally DNMT3A localises to H3K36me2, a mark broadly distributed in euchromatin [34] whereas DNMT3B is localised to the transcription-associated mark H3K36me3 in mouse embryonic stem cells [33] and colorectal cancer cells [32]. It is possible that the activity of these *de novo* methyltransferases compensates for any failure to maintain methylation at euchromatin by DNMT1. Normally such activity is low at heterochromatin and this inability to correct errors made by DNMT1 could underpin the formation of PMDs. This hypothesis is supported by our observation that the kinetics of DNA methylation loss in DNMT1 degron cells are consistent with lower *de novo* activity in PMDs relative to HMDs. It is also is supported by our observation that DNMT3A localises to hypermethylated PMDs in DNMT1 KO cells and a report that *Kras* mutant mouse lung cancers lacking DNMT3A have a more uniformly hypomethylated genome than those possessing DNMT3A [49]. Previously, we also observed that PWWP domain mutations can cause DNMT3B to re-localise to H3K9me3-marked heterochromatin also leading to the hypermethylation of PMDs [31]. Here we observe downregulation of DNMT3B2

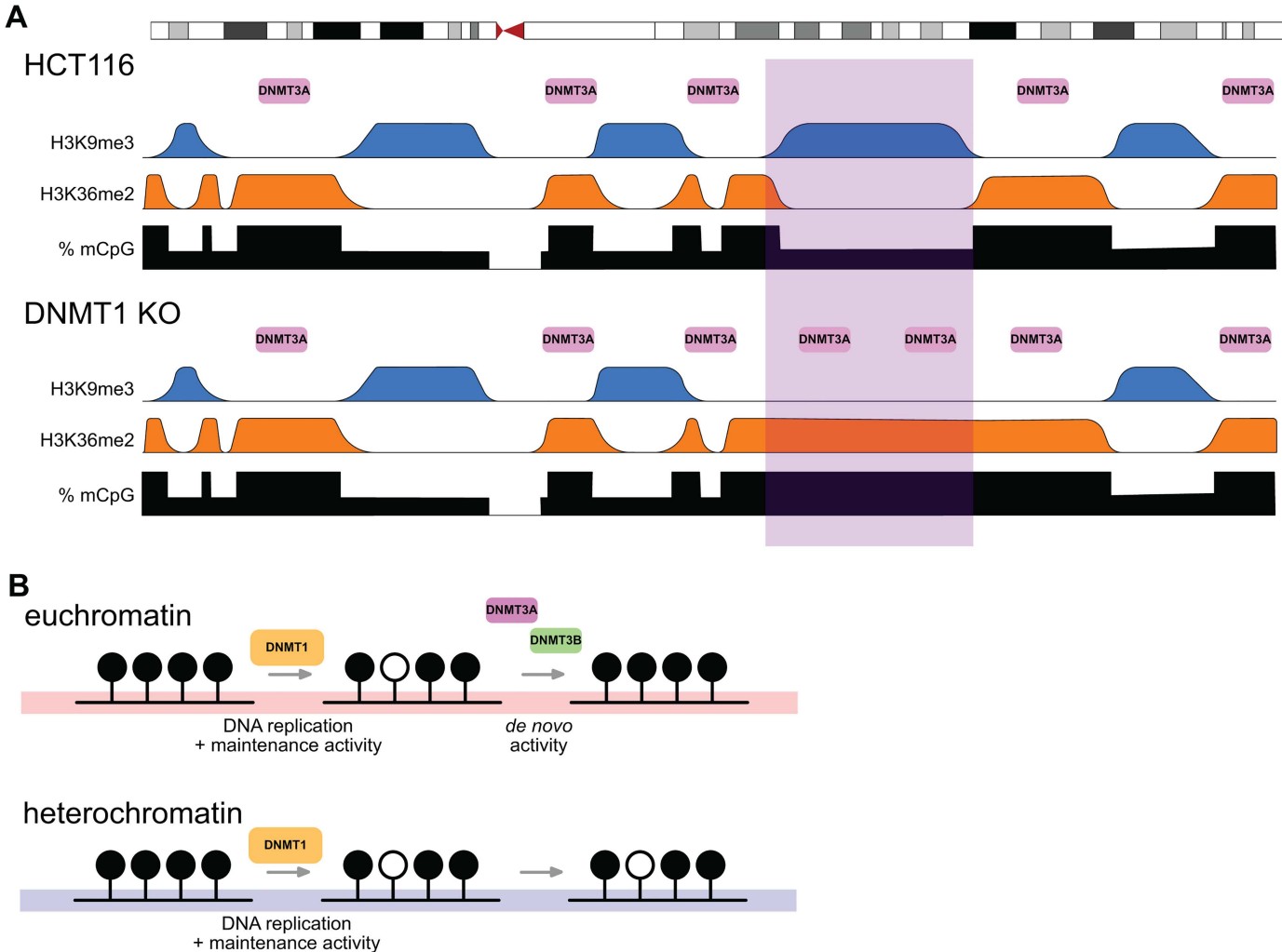

**Fig 7. DNMT1 loss leads to hypermethylation of a subset of late replicating domains by DNMT3A. (A)** Schematic showing distribution of DNA methylation, histone modifications and DNMT3A at the domain level in HCT116 and DNMT1 KO cells. DNMT3A predominantly localises to H3K36me2-marked regions through the action of its PWWP domain [34] and is excluded from heterochromatic domains. We propose that chromatin remodelling in DNMT1 KO cells leads to loss of heterochromatic H3K9me3 in some regions and gain of H3K36me2. This causes localisation of DNMT3A and gains of DNA methylation. **(B)** Schematic illustrating potential role of differential *de novo* DNMT activity in the formation of PMDs. Normally, DNMT3A and DNMT3B localise to euchromatic regions primarily through association with H3K36me2/3 [32–34]. This means that *de novo* DNMTs could correct mistakes made by DNMT1 following replication in euchromatin but not heterochromatin. This differential activity could therefore underpin the formation of PMDs at heterochromatin in cancer.

which could lead to a greater frequency of DNMT3A1-DNMT3B3 heterotetramers. Given that a recent study observed that DNMT3L expression resulted in methylation of PMDs by DNMT3B in human trophoblast stem cells [50], it is possible that the relative increase of DNMT3B3 may re-localise DNMT3A to heterochromatin. However, at present the influence of DNMT3B3 on DNMT3A genomic localisation remains unclear. Our observations suggest that previously reported differences in the re-methylation kinetics of different parts of the genome following DNA replication [20–22], could be caused by varying levels of *de novo* activity rather than variation in the kinetics by re-methylation by DNMT1 across the genome.

We observe that the localisation of DNMT3A to hypermethylated PMDs is associated with both a loss of H3K9me3 and a gain of H3K36me2. Previous work has observed that H3K9me3-marked heterochromatin excludes DNMT3A in brain

development [38]. In sperm development, heterochromatic regions gain DNA methylation upon loss of H3K9me3 and gain of H3K36me2 [37,50] suggesting that both marks contribute to DNMT3A localisation [51]. Our observations of recruitment of DNMT3A to hypermethylated PMDs when H3K9me3 is reduced and H3K36me2 is elevated are consistent with these reports and also the observations that differences in the locations of PMDs between different head and neck tumours correlate with differences in H3K36me2 patterning [52]. We and others have reported that DNMT3A is also recruited to the polycomb-associated H2AK119ub mark through its N-terminal region [53–57]. However, we do not observe the polycomb-associated H3K27me3 mark, which is generally tightly correlated with H2AK119ub [58], at hypermethylated PMDs. Furthermore, DNMT3A's localisation is predominantly driven by PWWP-dependent recruitment to H3K36me2 unless its PWWP domain is mutated [57], suggesting that H2AK119ub does not play a role in the recruitment of DNMT3A to hypermethylated PMDs. We note that instead H3K27me3 is lost from the boundaries of hypermethylated PMDs and that this loss could potentially play a role in their hypermethylation.

Taken together, our observations suggest that hypermethylated PMDs result from a reconfiguration of chromatin state in these regions during the generation of DNMT1 KO cells. They also suggest that DNMT3A re-localisation occurs following disruption of heterochromatin caused by DNMT1 loss. Indeed losses of H3K9me2/3 are previously reported in DNMT1 KO cells [59] and acute depletion or inhibition of DNMT1 reduces H3K9me3 levels and alters heterochromatin compartmentalisation [60,61]. This could allow H3K36 methyltransferases access and deposition of H3K36me2. Further evidence of interplay between these two marks is provided by another recent study demonstrating that perturbation of H3K36me2 leads to alterations in H3K9me3 distribution [40]. In support of this hypothesis, we find that re-expressing of full-length DNMT1 does not lead to a reversion of methylation levels at hypermethylated PMDs. The ultimate cause of this change is unclear, however, and cannot be dissected using a stable knockout system. Future studies using time-resolved models and manipulation of DNMT3A will be required to understand the interplay between DNMT1 loss, changes in chromatin structure and *de novo* DNMT localisation.

DNA sequence has also been proposed to play a central role in the formation of PMDs. Within PMDs single CpG methylation levels correlate with CpG density [62] and hypomethylation is particularly apparent at CpGs that are flanked by A or T nucleotides and distant from other CpGs, termed solo-WCGW CpGs [24]. Methylation levels at solo-WCGWs negatively correlate with the number of mutations in tumours [24] and also with population doublings *in vitro* [63]. A bias towards loss of methylation from low CpG density regions is also reported for DNMT1 depletion by shRNA [48]. Here, we find that a PMD can convert to a non-PMD in association with alterations histone modifications and recruitment of *de novo* DNMTs without a change in sequence. This suggests that whereas sequence may play a role in PMD formation, it can be superseded by differences in chromatin structure and DNMT recruitment. The observation that tissue- and tumour subtype-specific PMDs exist [17,52] also demonstrates that the formation of PMDs is not solely driven by DNA sequence.

DNMT1 knockout HCT116 cells were generated by deletion of 3 exons of DNMT1 [26]. It was later shown that these cells express low levels of a truncated DNMT1 protein [27,28]. As with most mammalian cells, complete genetic removal of DNMT1 from HCT116 cells is lethal [64]. This means that it is possible that the formation of hypermethylated PMDs is a gain-of-function effect of the truncated DNMT1 allele. However, to our knowledge, the truncated protein has not been reported to present any additional gain-of-function effects. In addition, here we present evidence that hypermethylated PMDs display different behaviour to other PMDs in DNMT1 degron cells and clinical colorectal tumours suggesting they are not a specific result of this trunctated, hypomorphic allele.

Here we have examined DNMT3A and DNMT3B localisation by performing ChIP-seq on ectopically expressed, tagged transgenes which we have previously shown are overexpressed relative to endogenous levels [32]. However, we have also demonstrated that our overexpressed DNMT3B recapitulates the localization of the endogenously tagged protein to the genome [31] and our results reflect the consensus from previous studies [33,34]. Furthermore, ectopically expressed tagged-DNMT3A with HESJAS-causing PWWP mutations re-localises to polycomb-marked loci in cells [57]. This mirrors gains in methylation at these loci *in vivo* in both HESJAS patients and mouse models [43,65]. The combined evidence

therefore strongly suggests that ectopically expressed, tagged DNMT3A and DNMT3B can be used to understand the localisation of the endogenous proteins. Given that ChIP-seq signal is affected by residence time [66], it is possible that our findings of increased DNMT3A ChIP-seq signal in hypermethylated PMDs could reflect altered residence time rather than increased DNMT3A localisation.

It is possible that DNMT localisation and function differ between normal and cancerous cells and thus our findings may not reflect the formation of PMDs during carcinogenesis. Indeed, differences between cancer cell line and tumour methylation patterns have previously been noted [67]. However, here we also observe that PMDs in HCT116 cells reflect the domain-level methylome of colorectal tumours. Previous work on the role of DNMTs in colorectal cancer has largely focused on DNMT3B which is reported to be overexpressed in 15–20% of cases, often as a result of amplification [68–71]. Its deletion or overexpression in mouse models also alters the rate of colorectal carcinogenesis [69,72,73] suggesting that it is a tumour promoting gene. DNMT3A expression in colorectal tumours is less studied but it is reported to be upregulated in at least some tumours [71,74,75]. In addition, DNMT3A deletion inhibits colorectal carcinogenesis in a mouse model [75] suggesting it plays a functional role. Here, we provide evidence that it plays a key role in the organisation of the colorectal cancer methylome providing a basis for future studies as to its role in colorectal carcinogenesis using pre-clinical models that better reflect this process than cancer cell lines.

In conclusion, we report an unexpected consequence of DNMT1 knockout in colorectal cancer cells is the gain of DNA methylation in several PMDs in association with the localisation of DNMT3A. These regions reveal a role of DNMT3A in maintaining DNA methylation homeostasis in cancer cells and, more generally, suggests that the lack of *de novo* activity in heterochromatin is a key factor in the formation of PMDs in cancer.

## Methods

### Cell culture

HCT116 and DNMT1 KO cells were gifts from B. Vogelstein [26]. Cells were cultured in McCoy's 5A (Gibco) supplemented with 10% Fetal Calf Serum (Life Technologies) and penicillin–streptomycin antibiotics at 140 and 400 µg/ml, respectively at 37°C with 5% $CO_2$. Cell lines were routinely tested for mycoplasma contamination. DNMT1 KO genotype was confirmed by western blot and sequencing.

### DNMT3A/B and DNMT1 expression in cells

Stable cell lines ectopically expressing DNMT3A or DNMT3B were generated for HCT116 cells and DNMT1 KO cells. A stable cell line ectopically expressing DNMT1 was generated for DNMT1 KO cells. For DNMT3A/B expression, PB-CAG-T7-DNMT3B2-IRES-puro or PB-CAG-T7-DNMT3A1-IRES-puro [31] were used. DNMT1 was cloned from a plasmid gifted from R. Meehan into PB-CGIP (a gift from M. McCrew) by swapping eGFP. All transfections were performed with a plasmid expressing piggyBac transposase using FuGene HD transfection reagent (Promega). After transfection, cells were grown in media supplemented with 2µg/ml puromycin to generate stably expressing cell lines.

### Western blotting

Whole-cell extracts were obtained by sonication in urea buffer (8M Urea, 50 mM Tris pH 7.5, 150mM β-mercaptoethanol). Extracts were analysed by SDS–polyacrylamide gel electrophoresis using 4–12% Bis-Tris NuPAGE protein gels (Life Technologies) and transferred onto a nitrocellulose membrane in 2.5mM Trisbase, 19.2 mM glycine and 20% methanol. Immunoblotting was performed following blocking in 10% Western blocking reagent (Roche) using antibodies against DNMT3B (Cell Signalling Technology, D7O7O, 1:1000), GAPDH (Cell Signalling Technology, 14C10, 1:6000 or Abcam ab125247), DNMT3A (Cell Signalling Technology, 2160, 1:500). Images were acquired with Odyssey DLx Imager following incubation with IRDYE 800CW goat anti-rabbit or IRDYE 680RD goat anti-mouse IgG secondary antibodies (LICORbio 1:20000). Uncropped Western blots for all replicates are provided in *S10 Fig*.

## DNA extraction

DNA was extracted from cell pellets snap-frozen in dry ice with ethanol. Cells pellets were resuspended in Genomic Lysis Buffer (300mM NaCl, 1% SDS, 20mM EDTA). The lysis mixture was syringed up and down using a 21G needle before being incubated at 55°C overnight with proteinase K. RNA was removed by incubation with RNase A/T1 Cocktail (Ambion) at 37 °C for 1 h, in between two phenol–chloroform extraction steps. DNA was quantified by Nanodrop 8000 spectrophotometer or Qubit fluorometer (Invitrogen) and purity was assessed using the Agilent 2100 BioAnalyzer.

## WGBS data generation

DNA (100ng) was bisulphite converted using the EZ DNA Methylation-Lightning kit, according to the manufacturer's instructions. Libraries were generated using the TruSeq DNA Methylation kit, according to the manufacturer's instructions. Sequencing was performed at the Edinburgh Clinical Research Facility. Libraries were sequenced using the Illumina NextSeq 500/550 High-Output v2 (150 cycles) (#FC-404–2002) on the NextSeq 550 platform (Illumina Inc, #SY-415–1002) over 1 flow cell.

## WGBS data processing

For WGBS generated as part of this study, Fastq files were quality checked using FastQC (v0.11.4) and reads trimmed with TrimGalore (v0.4.1) using default settings. Reads were then mapped to a bisulfite converted genome (hg38) and duplicates were removed using bismark (v 0.18.1) with bowtie2 (v2.3.1) for paired end data (settings: -N0 -L20) [76,77] before PCR duplicates were identified and removed using Bismark's deduplicate_bismark command. Aligned BAM files were processed to report coverage and the number of methylated reads for each CpG observed. Forward and reverse strands were combined using Bismark's methylation extractor and bismark2bedgraph modules with custom Python and AWK scripts. Processed WGBS files were assessed for conversion efficiency based on the proportion of methylated reads mapping to the phage-λ genome spike-in (>99.5% in all cases). For summary of WGBS alignment statistics, see *Table A in* S1 Tables.

To reanalyse published WGBS, processed data files were downloaded from NCBI GEO. AID-DNMT1 degron HCT116 cell WGBS was downloaded from series GSE236026 [35]. These files were processed to bedgraph format using custom scripts and weighted mean domain methylated was calculated as described below. Published tissue WGBS files from tissues were downloaded from series: GSE32399 (1 normal, 1 tumour) [9], GSE46644 (1 normal, 1 tumour) [47], GSE52271 (1 normal, 2 tumours) [43], GSE284325 (7 normal, 9 tumours) [44] and GSE212391 (10 tumours) [46] after filtering for samples representing normal colon or colorectal tumour samples. These processed data were universally processed to hg38 bedgraph format files using custom scripts and the UCSC liftoever tool before weighted mean domain methylation was calculated as described below.

Weighted mean methylation levels over genomic windows and domains were calculated by intersecting each window/domain with CpGs using bedtools *intersect* and *groupby* functions (v2.27.1, HCT116 and DNMT1 KO WGBS) or using the *bedtools* map function (v2.31.1, all other analyses).

Analysis of kinetics of de-methylation for AID-DNMT1 degron cells was conducted using linear models in R. Models were fitted to log weighted mean methylation values for each domain using the day 0–6 timepoints where slopes was observed to be approximately linear before deriving the estimate slope for each domain. Final levels were taken as the day 12 weighted mean values.

## PMD annotation

To define PMDs, we used Methpipe (v5.0.0) (Decato et al., 2020) on CpG level WGBS data. Methpipe defined PMDs were further processed by trimming the domains obtained to the first and last CpG observed withing the domains, as well as by

bridging nearby domains whose distance was smaller than 2x the window size used. The rest of the genome, not anno-tated as PMDs, was then assigned as highly methylated domains (HMDs) using bedtools (v2.27.1). We excluded poorly mapped regions of the genome using annotations of gaps and centromeres (using hg38 gap and centromere tracks down-loaded from UCSC table browser). Regions annotated as heterochromatin, short arm and telomeres from the gaps track were merged with the centromeres track using bedtools merge with -d set to 10Mb. This merged file was then excluded from the PMD and HMD BED files using bedtools subtract. Annotated domains smaller than 200kb were also removed. After removal of the < 200kb domains, the neighboring domains were then merged. Hypermethylated PMDs were defined as those ≥ 5% mean methylation in DNMT1 KO versus HCT116 cells.

### Pileup plots

ComputeMatrix from Deeptools (v3.5.0) was used to generate pileup plots and heatmaps. All domains were normalised to a size of 1Mb based on their start and end points using a window size of 10Kb (--binsize 10000 (-m 1000000). The flanks were plotted using the same window size and a maximum distance of 1Mb -b 1000000 -a 1000000).

### Repli-seq data generation

Repli-seq data was generated from cells at approximately 70% confluency to ensure most of the cells were actively cycling. To label DNA, cells were given fresh media supplemented with 100µM 5-Ethynyl-2'-deoxyuridine (EdU) and incu-bated at 37°C for 30 minutes. Cells were then collected by trypsinization along with media and a PBS wash of the flask to ensure non-adherent mitotic cells were not lost. The cell suspension was then centrifuged at 200g for 5 minutes and washed with 10ml PBS before being resuspended in 2.5ml PBS supplemented with 1% FCS. Cells were fixed by gently adding 7.5ml of 100% ethanol to a final concentration of 70% with gentle vortexing followed by incubation on ice for 30 mins and storage at –20°C.

Fixed cells were washed with 5ml ice cold PBS and then resuspended in 5ml 100mM HCl and 0.015% pepsin (0.75mgr) in $H_2O$ before being incubated at 37°C for 30 minutes while rotating. Following centrifugation at 600g for 10 minutes, pellets were washed with 5 ml ice cold PBS and then resuspended in 1ml PBS along with 50µl Propidium Iodide (PI) and 20µl RNase A and incubated at room temperature for 30 minutes while protected from light. Extracted Nuclei were counted using a hematocytometer and suspended to approximately $3x10^6$ nuclei/ml to ensure optimal flow during FACS sorting. Nuclei were sorted on a FACSAria flow cytometer (BD Biosciences) in PBS supplemented with 0.25% BSA based on their DNA content using PI staining, into three fractions: Early, Middle and Late S-Phase (defined by equally splitting the window between the G1 and G2 peaks into three equal portions).

DNA was then extracted from sorted nuclei as described above except they were sonicated on a Bioruptor following proteinase K digestion (15 cycles of 30 seconds on – 30 seconds off, on high at 4°C). Precipitation of extracted DNA was then performed with linear acrylamide (LPA) carrier and isopropanol and resuspended in TE. DNA was then quantified using Qubit High Sensitivity reagents and sonicated on a Covaris E220 sonicator to yield 100–300 bp fragments which were quality controlled on a Bioanalyser.

Re-precipitation of DNA was then performed with LPA carrier and ethanol before being resuspended in water. A click reaction was then performed to conjugate biotin to the EdU by adding 3ul 2M TAB (0.4M), 30ul DMSO, 6ul 5mM Ascorbic Acid (2mM), 1.2ul 10mM Biotin Azide (0.8mM), 3ul 10mM CuTBTA (2mM) and 1.8ul $H_2O$. Samples were then incubated overnight at room temperature while covered from light and cleaned up by ethanol precipitation with LPA carrier.

EdU containing DNA was then enriched using Myone C1 Streptavidin Dynabeads (Invitrogen) and a magnetic rack overnight at 4°C with rotation. Following this, the samples were washed at 4°C with: TSE-I (20 mM Tris-HCl, pH 8.1, 2 mM EDTA, 150 mM NaCl, 1% Triton, 0.1% (v/v) SDS) x2, TSE-II (20 mM Tris-HCl pH 8.1, 2 mM EDTA, 500 mM NaCl, 1% Triton, 0.1%(v/v) SDS) x1 and TE x1. Samples were eluted by adding water at 95°C and incubation with vortexing at 95°C for 10 mins. Successful enrichment of early and late replicating DNA was then validated by qPCR of early (BMP1, TEK,

MELK, RTTN) and late replicating (NETO1, CDH8, DPPA2 and PTPRD) loci, based on a published protocol and HCT116 WT repli-seq data [78,79]. Primer sequences used in this validation are included in *Table B in S1 Tables*.

Repli-seq libraries were then prepared using the Acel-NGS 1a Plus DNA library kit along with the 1A Plus Set A Indexing Kit following the manufacturer's instruction (Swift Biosciences). However, as our DNA template contain Uracil due to the EdU incorporation, the kit's polymerase, and subsequent buffer (Reagent W2, Buffer W3 and Enzyme W4) were replaced with KAPA HiFi HotStart Uracil+ Ready Mix Kit (Roche), which included an uracil tolerant polymerase. Libraries were assessed for size distribution on the Agilent Bioanalyser (Agilent Technologies) and quantified using the Qubit 2.0 Fluorometer. Repli-seq libraries were sent for whole genome sequencing at the Wellcome Trust Clinical Research Facility (WTCRF). The libraries were sequenced using the Illumina NextSeq 500/550 High-Output v2.5 (75 cycles) Kit on the Illumina NextSeq 550 platform over 2 flow cells to give 75 bp single-end reads.

## Repli-seq data processing

Repli-seq fastq files were quality checked using FastQC (v0.11.4) and reads were trimmed using TrimGalore (v0.4.1) using default settings. Reads were then mapped to the genome (hg38) using bowtie2 for paired end data (version settings: -N1 -L 20 --no-unal -- no-mixed -- no-discordant -X 1000) [77]. Low mapping quality reads were removed using samtools (v1.6, settings: -bq10). Repli-seq data were analysed as previously described [80]. Briefly, read counts were generated for 10kb genomic windows using bedtools coverage (v2.27.1, settings: -counts) and normalised to reads per million (RPM). Replication timing was then calculated as the ratio between the early and late fraction RPM counts. Quantile normalisation of counts was then performed across samples and Loess smoothing was performed throughout all chromosomes using R (expect for the Y chromosome and mitochondrial DNA). The RPM counts over 10kb windows for early, middle, and late samples as well as the Loess smoothed replication timing counts were converted into BigWigs using bedGraphToBigWig from UCSC tools. For a summary of Repli-seq alignment statistics, see *Table C in S1 Tables*.

## ChIP data generation

ChIP-seq was performed as previously described [31,32]. For T7-DNMT3B and T7-DNMT3A ChIP-seq experiments, $1 \times 10^7$ cells were harvested, washed and crosslinked with 1% methanol-free formaldehyde in PBS for 8 min at room temperature. Crosslinked cells were lysed for 10 min on ice in 50 μl of lysis buffer (50 mM Tris-HCl pH 8, 150 mM NaCl, 1 mM EDTA, 1% SDS) freshly supplemented with proteinase inhibitor (Sigma-Aldrich). IP dilution buffer (20 mM Tris-HCl pH 8, 150 mM NaCl, 1 mM EDTA, 0.1% Triton X-100) freshly supplemented with proteinase inhibitor, DTT and PMSF was added to the samples to reach a final volume of 500 μl. Chromatin was fragmented using Benzonase [81]: samples were sonicated on ice with Soniprep 150 twice for 30 s to break up nuclei; then 200 U of Benzonase Nuclease (Sigma) and MgCl2 (final concentration 2.5 mM) were added and samples were incubated on ice for 15 min. The reaction was blocked by adding 10 μl of 0.5 M EDTA pH 8. Following centrifugation for 30 min at 18,407 g at 4 °C, supernatants were collected and supplemented with Triton X-100 (final concentration 1%) and 5% input aliquots were retained for later use. Protein A Dynabeads (Invitrogen) previously coupled with 10 μl of T7-Tag antibody per $1 \times 10^7$ cells in blocking solution (1x PBS, 0.5% BSA) were added and the samples incubated overnight under rotation at 4 °C. Beads were then washed for 10 min at 4 °C with the following buffers: IP dilution buffer 1% Triton X-100 (20 mM Tris-HCl pH 8, 150 mM NaCl, 2 mM EDTA, 1% Triton X-100), buffer A (50mM HEPES pH 7.9, 500 mM NaCl, 1 mM EDTA, 1% Triton X-100, 0.1% Na-deoxycholate, 0.1% SDS), buffer B (20 mM Tris pH 8, 1 mM EDTA, 250 mM LiCl, 0.5% NP-40, 0.5% Na-deoxycholate), TE buffer (1mM EDTA pH 8, 10 mM Tris pH 8). Chromatin was eluted by incubating the beads in extraction buffer (0.1 M NaHCO3, 1% SDS) for 15 min at 37 °C. To reverse the cross-linking Tris-HCl pH 6.8 and NaCl were added to final concentrations of 130 mM and 300 mM respectively, and immunoprecipitations were incubated at 65 °C overnight. Samples were then incubated at 37 °C for 1 h after addition of 2 μl of RNase Cocktail Enzyme Mix (Ambion). Then 40 μg of Proteinase K (Roche) were added, followed by 2 h incubation at 55 °C. Input material was similarly de-crosslinked. Samples were purified with the MinElute

PCR purification kit (QIAGEN). For ChIP-Rx-seq of ectopic T7-DNMT3B and T7-DNMT3A 20 µg of Spike-in chromatin (ActiveMotif 53083) was added to each sample after sonication. 2 µl of spike-in antibody per sample (ActiveMotif 61686) was also added in a ratio 1:5 versus the T7 antibody. A similar protocol was used for H3K4me3, H3K9me3, H3K27me3 and H3K36me3 ChIP-seq experiments, except: $0.5 \times 10^7$ cells were harvested and crosslinked with 1% methanol-free formaldehyde in PBS for 5 min at room temperature. For H3K36me3 ChIP-Rx-seq crosslinked Drosophila S2 cells were spiked into samples before sonication at a ratio of 20:1 human to Drosophila cells. Following nuclei rupture by sonication on ice with Soniprep 150, chromatin was fragmented using Bioruptor Plus sonicator (Diagenode) for 40 cycles (30 s on/30 s off on high setting at 4 °C). $2 \, \mu l / 1 \times 10^6$ cells of the following antibodies were used for immunoprecipitations: H3K4me3 (EpiCypher 13–00041), H3K9me3 (Active Motif 39161), H3K27me3 (Cell Signaling Technology C36B11) and H3K36me3 (Abcam ab9050).

For H3K36me2, native ChIP-seq was performed adapting ultra-low-input (ULI) ChIP [82] as follows. Nuclei were isolated from $1 \times 10^6$ unfixed cells using PBS 0.1% NP-40 and resuspended in 75 µl of Nuclear Isolation buffer (10mM Tris HCl, 0.1% NaDeOx, 0.1% Triton-100) and 75 µl of 2X MNase digestion Buffer (NEB). Chromatin was digested using MNase (NEB) for 6 min at 37 °C. Enzyme concentration was determined experimentally to obtain a majority of mononucleosomes and reaction stopped adding 5 µl of 0.5 M EDTA pH 8. Nuclei were lysed as follows: 30 µl of 5X Nuclear Lysis Buffer (5% NaDeOx, 5% Triton X-100) were added and samples vortexed, then 160 µl of ChIP Dilution buffer (0.1% Triton X-100, 20 mM Tris-HCl (pH 8.1), 150mM NaCl) were added and samples sonicated using Bioruptor Plus sonicator (Diagenode) for two cycles (30 sec on/30 sec off on high setting at 4 °C). Following centrifugation for 30 min at maximum speed at 4 °C, supernatants were collected and 350 µl of ChIP Dilution buffer added. Half the sample was retained to check chromatin fragmentation and half processed for immunoprecipitation as previously described. 10% of input retained and antibody-bound-beads added for overnight incubation at 4 °C. 3 µl of anti-H3K36me2 (Invitrogen, T.571.7, MA5–14867) were used per sample. Washes and DNA elution were performed as described above.

Prior to library generation, we performed qPCR using SNAP-ChIP synthetic modified nucleosomes to validate antibodies for H3K4me3, H3K9me3, H3K27me3 and H3K36me2. We also added these nucleosomes into samples and performed qPCR to check specificity and enrichment of the antibodies for each sample and checked a positive and negative locus for the marks H3K4me3, H3K27me3 and H3K9me3 based on ENCODE data for HCT116 cells [83]. Primers against MELK were used as a positive locus for H3K4me3 and as a negative locus for H3K9me3 and H3K27me3. Primers against ELAVL2 were used as a positive locus for H3K9me3 and as a negative locus for H3K4me3. Primers against MLLT3 were used as a positive locus for H3K27me3. Validation of successful DNMT-ChIP by qPCR was performed with a positive and negative locus for DNMT3A and DNMT3B [32]. Primers against DAZL were used as a positive locus for DNMT3A, and against TNFRSF1A for DNMT3B. Primers against BRCA2 were used as a negative locus for both DNMT3A and DNMT3B. Primer sequences used in this validation are included in *Table B in S1 Tables*.

ChIP-seq libraries were prepared using the NEBNext Ultra II DNA Library Prep Kit for Illumina (NEB) and NEBNext Multiplex Oligos for Illumina (NEB) barcode adapters according to the manufacturer instructions. were used. Specifically, Illumina Index Primers Set 1 (E7335) for H3K36me3, and Unique Dual Index UMI Adaptors DNA Set 1 (E7395) for the other libraries. For histone modifications ChIP-seq, adapter-ligated DNA was size selected for an insert size of 150 bp using Agencourt AMPure XP beads. Libraries were quantified using the Qubit dsDNA HS or BR assay kit and assessed for size and quality using the Agilent Bioanalyser. H3K36me3 ChIP-Rx-seq libraries were sequenced using the NextSeq 500/550 high-output version 2.5 kit (75 bp single end reads). The other ChIP-seq libraries were sequenced using the NextSeq 2000 P3 (50 bp paired end reads). Libraries were combined into equimolar pools to run within individual flow cells. Sequencing was performed at the Edinburgh Clinical Research Facility. HCT116 ChIP-seq data for H3K4me3, H3K9me3, H3K27me3 and H3K36me3 used in this study have previously been reported [31,32], and DNMT1 KO data were generated in parallel.

## ChIP-seq data processing

Fastq files were quality checked using FastQC (v0.11.4) and reads trimmed using TrimGalore (v0.4.1). Reads were then mapped to the genome (hg38) using bowtie2 (v2.3.1 with settings: -N 1 -L 20 --no-unal) [77] for paired end data. Low mapping quality reads or fragments were removed using samtools (v1.6 with settings -bq 10). Duplicate reads were removed using sambamba markdup (v0.5.9, for H3K4me3, H3K9me3, H3K27me3 and H3K36me3 ChIP-seq) or using UMI tools dedup function (v1.0.0 and setting: --paired, DNMT3A/B and H3K36me2 ChIP-seq). The number of reads pre- and post- alignment as well as the reads after deduplication and low mapping was then extracted and counted. Bed counts for 10 kb genomic windows and domains were generated from the bam files using bedtools coverage (seetings: -*counts*, v2.27.1). For paired-end data, the BAM file was first converted to a BED file of fragment locations using BEDtools bamtobed function. Coverage counts were scaled to counts per 10 million based on total number of mapped reads per sample and divided by the input read count to obtain a normalised read count. An offset of 0.5 was added to all windows prior to scaling and input normalisation to prevent intervals with zero reads in the input sample generating a normalised count of infinity. Regions where coverage was 0 in all samples were removed from the analysis. BigWig files for visualisation in the genome browser were generated, by calculating counts per million mapped reads using bamCoverage from deeptools (v3.2.0). Normalisation over inputs and means between replicas were then performed using bigwigCompare ratio and mean respectively. For a summary of ChIP-seq alignment statistics, see *Tables D to I in S1 Tables*.

## Definition of H3K9me3-marked and H3K27me3-marked PMDs

Analysis of H3K9me3 and H3K27me3 enrichment across PMDs, showed a bimodal pattern for both marks, where high enrichment H3K9me3 PMDs showed low H3K27me3 enrichment and vice versa. K-means clustering was used to divide the PMDs into 2 clusters of H3K9me3-marked PMDs and H3K27me3-marked PMDs.

## Definition of H3K9me3 domains

H3K9me3 and H3K27me3 domains were called using a hidden Markov model as previously described [16]. Normalised mean H3K9me3 ChIP-seq coverage in 25 kb genomic windows was analysed using the bigwig_hmm.py script (https://github.com/gspracklin/hmm_bigwigs) to define two states (-n 2). We excluded poorly mapped regions of the genome from these domains using annotations of gaps and centromeres from the UCSC browser (hg38 gap and centromere tracks). Annotations were downloaded from the UCSC table browser. Regions annotated as heterochromatin, short arm and telomeres from the gaps track were merged with the centromeres track using BEDtools merge with -d set to 10 Mb. This merged file was then excluded from the domains BED files using BEDtools subtract.

## RNA-seq data generation

RNA extraction was performed from snap-frozen cell pellets using RNeasy Plus Mini Kit (QIAGEN) following the manufacturer's instructions. Quantity and quality of RNA samples were assessed using the Nanodrop spectrophotometer (Nanodrop ND-1000, Thermo Scientific) and Qubit fluorometer (Invitrogen). Size distribution of RNA fragments and the RNA integrity number (RIN) value were determined by using Agilent 2100 BioAnalyzer with the RNA nano chip. RNA samples for whole genome sequencing were sent to Edinburgh Genomics for library preparation and sequence data generation. Library preparation was performed using the TruSeq stranded mRNA-seq library kit. The libraries were then sequenced on NovaSeq with 50 bp paired end (PE) reads and aiming for 50M reads per sample.

## RNA-seq data processing and analysis

RNA-seq fastq files were quality checked using FastQC (v0.11.4) and reads were trimmed using TrimGalore (v0.4.1) and default settings. Reads were then mapped to the genome (hg38) using bowtie2 for paired end data (2.3.1, settings: -N1 -L

20 --no-unal -- no-mixed -- no-discordant -X 1000) [77]. Low mapping quality reads were removed using samtools (settings: -bq10). The bam file was then converted into bed using bedtools and CPM values calculated. BigWig files were then generated using the ucsc toolset, for visualisation in the genome browser. FeatureCounts (settings: -T 4 -t exon -p, from subread package) was then used to count how many reads align to each gene. For a summary of RNA-seq alignment statistics, see *Table J in S1 Tables*. Part of the HCT116 RNA-seq data used here was previously reported [31] and the rest of the data was generated in parallel.

Statistical analysis to detect differentially expressed genes between genotypes was performed using the edgeR package in R [84]. Normalisation factors were then calculated using the trimmed mean of M values (TMM) normalisation method and differential expression analysis was performed using a general linear model (GLM). Statistically significant differentially expressed genes were defined as the genes that demonstrated a false discovery rate (FDR) < 0.05 and an absolute log fold change > 1. For this analysis lowly expressed genes across all samples were excluded using the EdgeR *filterByExpr* command with default parameters. To compute changes in gene expression for all genes within domains, the analysis was repeated without this exclusion and log fold changes extracted.

## Oxford Nanopore data generation and processing

DNMT1 rescue sample cell lines were sequenced on a Promethion 10.4.1 flow cell. DNA samples were sheared to approximately 20kbp using the Megaruptor 3, end-repaired and then barcoded with the Native Barcoding Kit 24 V14 (Oxford Nanopore Technologies, #SQK-NBD114.24). Barcoded DNA samples were combined in a single pool prior to sequencing adapter ligation. 100fmol of sequencing library was loaded on a single R10.4.1 flow cell and sequenced on the PromethION 24 for 100 hours with washing/reloading after 24 and 72 hours.

Basecalling was then performed using high-accuracy model v4.3.0, 400 bps to yield methyl- and hydroxymethyl- cytosine calls in CpG contexts only. Resulting BAM files of reads were aligned to hg38 using Dorado (v0.7.2) with integrated Minimap2-2.28 (r1209, default settings). Methylation data was extracted at reference CpG sites using modkit pileup (v0.3.1, settings: --cpg, --filter-threshold C:0.78). Weighted mean methylation values were then computed using methyl-cytosine calls and the modified plus valid coverage columns by converting modkit bedmethyl files to bedgraph files and conducting downstream analyses similarly to WGBS.

Published processed Nanopore methylation calls from 22 colorectal tumours were downloaded from NCBI GEO GSE270257 [45]. These were then converted to bedgraph files and handled identically to published WGBS described above.

## Annotation of functional elements

Gene positions in genome browser plots are hg38 RefSeq genes taken from UCSC. CpG island positions were taken from Illingworth *et al*, converted to hg38 using the UCSC liftover tool [85].

## Statistical analysis

Statistical tests were performed in R (3.3.3) or GraphPad Prism (9.0.0).

## Supporting information

**S1 Fig. Ablation of DNMT1 leads to preferential hypomethylation of partially methylated domains.** (**A**) Boxplot showing replication timing of HCT116 PMDs (n = 546 domains) and HMDs (n = 558 domains). Replication timing data are mean loess smoothed repli-seq early/late ratios over 10 kb. (**B**) Boxplot showing H3K9me3 levels of HCT116 PMDs (n = 546 domains) and HMDs (n = 558 domains). ChIP-seq data are mean normalised IP/IN. (**C**) Boxplot showing H3K27me3 levels of HCT116 PMDs (n = 546 domains) and HMDs (n = 558 domains). ChIP-seq data are mean normalised

IP/IN. (**D**) Representative genomic locus showing histone marks at genes in HCT116 cells. Genome browser plot showing DNA methylation levels (mC) alongside HCT116 histone modification ChIP-seq and gene expression. DNA methylation levels are plotted for individual CpGs with coverage ≥ 5. ChIP-seq are normalised $\log_{10}$ IP/IN. RNA-seq are mean logRPKM. CGI = CpG islands. Chromosomal co-ordinates shown are hg38 chr9:35,000,000–35,150,000 (**E**) Heatmaps and pileup plots of HCT116 H3K9me3, H3K27me3 and DNA methylation levels alongside replication timing for H3K9me3-marked PMDs (n = 253 domains) and H3K27me3-marked PMDs (n = 293 domains). ChIP-seq data are mean normalised IP/IN, DNA methylation levels are mean % mCpG over 10kb. Replication timing data are mean loess smoothed repli-seq early/late ratios over 10kb. PMDs are aligned and scaled to the start and end points of each domain and ranked based on their mean methylation levels in HCT116 cells. (**F**) Violin plot showing mean HCT116 DNA methylation levels at H3K9me3 PMDs (n = 253 domains), H3K27me3 PMDs (n = 293 domains) and HMDs (n = 558 domains). (**G**) Boxplot showing replication timing of HCT116 H3K9me3 PMDs (n = 253 domains), H3K27me3 PMDs (n = 293 domains) and HMDs (n = 558 domains). Replication timing data are mean loess smoothed repli-seq early/late ratios over 10 kb. (**H**) Total DNA methylation levels are reduced in DNMT1 KO cells. Barplot of total methylated CpG levels estimated by WGBS. For boxplots: Lines = median; box = 25th–75th percentile; whiskers = 1.5 × interquartile range from box. All p-values are from two-sided Wilcoxon rank sum tests. All histone ChIP-seq and repli-seq data shown are derived from the mean of two biological replicates.
(PDF)

**S2 Fig. A subset of H3K9me3-marked PMDs are hypermethylated in DNMT1 knockout cells.** (**A**) Scatter plot of mean methylation levels at PMDs in DNMT1 KO cells versus HCT116 cells highlighting hypermethylated PMDs (hyper-PMDs). (**B**) Ideogram showing genomic distribution and size of hypermethylated PMDs. (**C**) Boxplot showing HCT116 H3K9me3 levels at hypermethylated PMDs (n = 60 domains) compared to other PMDs (n = 486 domains) and HMDs (n = 558 domains). ChIP-seq data are mean normalised IP/IN. (**D**) Boxplot showing HCT116 H3K27me3 levels at hypermethylated PMDs (n = 60 Domains compared to other PMDs (n = 486 domains) and HMDs (n = 558 domains). ChIP-seq data are mean normalised IP/IN. (**E**) Boxplot showing HCT116 replication timing at hypermethylated PMDs (n = 60 domains) compared to other PMDs (n = 486 domains) and HMDs (n = 558 domains). Replication timing data are mean loess smoothed repli-seq early/late ratios over 10kb. For boxplots: Lines = median; box = 25th–75th percentile; whiskers = 1.5 × interquartile range from box. All p-values are from two-sided Wilcoxon rank sum tests. All histone ChIP-seq and repli-seq data shown are derived from the mean of two biological replicates.
(PDF)

**S3 Fig. The replication timing of hypermethylated PMDs remains similar in DNMT1 KO cells.** (**A**) Density scatter plot showing genome-wide correlation between replication timing in DNMT1 KO and HCT116 cells. Replication timing data are mean loess smoothed repli-seq early/late ratios in 10kb windows. Spearman's correlation (Rho) and associated p-value is shown. (**B**) Boxplot showing HCT116 replication timing in 10 kb genomic windows divided in deciles according to their mean DNA methylation levels in HCT116. Replication timing data are mean loess smoothed repli-seq early/late ratios over 10 kb. Spearman's correlation coefficient (Rho) is shown alongside its associated p-value and n is the number of windows analysed. (**C**) Boxplot showing DNMT1 KO replication timing in 10 kb genomic windows divided in deciles according to their mean DNA methylation levels in DNMT1 KO. Replication timing data are mean loess smoothed repli-seq early/late ratios over 10 kb. Spearman's correlation coefficient (Rho) is shown alongside its associated p-value and n is the number of windows analysed. (**D**) Boxplot showing replication timing of HCT116 PMDs (n = 486 domains) and HMDs (n = 558 domains). Replication timing data are mean loess smoothed repli-seq early/late ratios over 10 kb. For boxplots: Lines = median; box = 25th–75th percentile; whiskers = 1.5 × interquartile range from box. All p-values are from two-sided Wilcoxon rank sum tests. All repli-seq data shown are derived from the mean of two biological replicates.
(PDF)

**S4 Fig. DNMT3A localises to hypermethylated PMDs.** (**A**) Volcano-plot showing differential expression of protein coding genes between HCT116 and DNMT1KO cells. DNMT1, DNMT3A and DNMT3B are indicated. FC = fold change. (**B,C**) Western blots of DNMT3A (**B**) and DNMT3B (**C**) protein levels in HCT116 and DNMT1 KO cells. In both, Western blots are a representative example. Bar heights indicate mean levels in protein extracts from 3 independent cell cultures normalised to HCT116 cells. Individual points indicate the level of each replicate. DNMT3BKO = DNMT3B knockout HCT116 cells to confirm antibody specificity. Uncropped blots are provided in S10 Fig. (**D**) Representative genomic loci showing DNMT3A/B localisation at a H3K9me3-marked PMD and an HMD in DNMT1 KO cells. Genome browser plots showing DNA methylation levels (mCpG) alongside DNMT3A/B ChIP-seq and HCT116 histone modifications and repli-seq. DNA methylation levels are plotted in 10 kb genomic windows. ChIP-seq tracks are normalised $\log_{10}$ IP/IN. Replication timing data are loess smoothed repli-seq early/late ratios over 10 kb. Representative H3K9me3-marked PMD and HMD are indicated by the coloured boxes. CGI = CpG islands. Chromosomal co-ordinates shown are from hg38, left: chr9:4,000,000–16,000,000 and right: chr9:31,000,000–40,000,000. (**E**) Boxplot comparing DNMT3A levels at PMDs (n = 546 domains) and HMDs (n = 558 domains). ChIP-seq data are mean normalised IP/IN. P-value from two-sided Wilcoxon rank sum test. (**F**) Boxplot comparing DNMT3B levels at PMDs (n = 546 domains) and HMDs (n = 558 domains). ChIP-seq data are mean normalised IP/IN. P-value from two-sided Wilcoxon rank sum test. (**G**) Boxplot showing mean HCT116 DNMTB levels in 10 kb genomic windows divided in deciles according to their mean levels of H3K36me3 in HCT116 cells. ChIP-seq data are normalised IP/IN. Spearman's correlation coefficient (Rho) is shown alongside its associated p-value and n is the number of windows analysed. (**H**) Boxplot showing mean HCT116 DNMT3A levels in 10 kb genomic windows divided in deciles according to their mean levels of H3K36me3 in HCT116 cells. ChIP-seq data are normalised IP/IN. Spearman's correlation coefficient (Rho) is shown alongside its associated p-value and n is the number of windows analysed. (**I**) Density scatter plot showing genome-wide correlation DNMT3A levels in DNMT1 KO and HCT116 cells. ChIP-seq data are normalised log IP/IN in 10kb windows. Spearman's correlation (Rho) and associated p-value is shown. (**J**) Density scatter plot showing genome-wide correlation DNMT3B levels in DNMT1 KO and HCT116 cells. ChIP-seq data are normalised log IP/IN in 10kb windows. Spearman's correlation (Rho) and associated p-value is shown. For boxplots: Lines = median; box = 25th–75th percentile; whiskers = 1.5 × interquartile range from box. All histone ChIP-seq and repli-seq data shown are derived from the mean of two biological replicates.
(PDF)

**S5 Fig. Hypermethylated PMDs lose DNA methylation more slowly following DNMT1 removal.** (**A**) Schematics illustrating predicted trajectories of DNA methylation levels following DNMT1 removal without (left) and with (right) differential *de novo* methyltransferase activity. Without *de novo* DNMT activity both will show exponential decay with an equivalent slope in log space (left). *De novo* methyltransferase activity will alter both the rate of decay and final plateaux level with higher *de novo* DNMT activity leading to slower loss and a higher final level. (**B**) Boxplots of mean methylation level for PMDs and HMDs following degradation of DNMT1 in AID-DNMT1 cells. (**C**) Boxplots of rate of loss of DNA methylation (left) and final levels of DNA methylation (right) in AID-DNMT1 cells for PMDs and HMDs. (**D**) Boxplots of rate of loss of DNA methylation (left) and final levels of DNA methylation (right) in AID-DNMT1 cells for H3K9me3-marked and hypermethylated PMDs. In plots, n = 558 domains for HMDs, 546 domains for PMDs, 210 domains for H3K9me3-marked PMDs and 60 for hypermethylated PMDs. For boxplots, lines = median; box = 25th–75th percentile; whiskers = 1.5 × interquartile range from box. All p-values are from two-sided Wilcoxon rank sum tests.
(PDF)

**S6 Fig. Hypermethylated PMDs lose H3K9me3.** (**A**) Representative genomic loci showing H3K9me3 and H3K36me2 levels at a representative H3K9me3-marked PMD and an HMD in DNMT1 KO cells. Genome browser plots showing DNA methylation levels (mCpG) alongside H3K9me3, H3K27me3 and H3K36me2 ChIP-seq. DNA methylation levels are plotted in 10 kb genomic windows. ChIP-seq tracks are normalised $\log_{10}$ IP/IN. Representative hypermethylated PMD is

indicated by the coloured box. CGI = CpG islands. Chromosomal co-ordinates shown are from hg38: chr9:18,000,000–30,000,000. (**B, C**) Density scatter plots showing genome-wide correlation H3K9me3 (**B**) and H3K27me3 (**C**) levels in DNMT1 KO and HCT116 cells. ChIP-seq data are normalised log IP/IN in 10kb windows. Spearman's correlations (Rho) and associated p-values are shown. (**D**) Heatmaps and pileup plots of HCT116 and DNMT1 KO DNA methylation levels alongside H3K27me3 levels for hypermethylated PMDs (n = 60 domains) and all other H3K9me3-marked PMDs (n = 253 domains). ChIP-seq data are mean normalised log IP/IN. DNA methylation levels are mean % mCpG. PMDs are aligned and scaled to the start and end points of each domain and ranked based on their mean methylation levels in HCT116 cells. All histone ChIP-seq data shown are derived from the mean of two biological replicates.
(PDF)

**S7 Fig. Hypermethylated PMDs gain H3K36me2.** (**A**) Density scatter plot showing genome-wide correlation H3K36me2 levels in DNMT1 KO and HCT116 cells. (**B**) Density scatter plots showing genome-wide correlation between H3K27me3 and H3K36me2 levels in HCT116 (left) and DNMT1 KO cells (left). (**C**) Density scatter plots showing genome-wide correlation between H3K9me3 and H3K36me2 levels in HCT116 (left) and DNMT1 KO cells (left). ChIP-seq data are normalised IP/IN in 10kb windows. In (**A-C**) Spearman's correlations (Rho) and associated p-values are shown. All histone ChIP-seq data shown are derived from the mean of two biological replicates. (**D**) Barplot of RNA-seq derived expression values for known H3K36 methyltransferses in HCT116 and DNMT1 KO cells. Expression values are shown mean Counts Per Million (CPM) calculated form n = 9 and 4 independent cultures of HCT116 and DNMT1 KO cells respectively. P-values shown are Benjami-Hochberg adjusted p-values derived from F-tests.
(PDF)

**S8 Fig. DNMT1 rescue leads to gains of methylation throughout the genome.** (**A**) Barplot of total CpG DNA methylation levels estimated by Nanopore sequencing from all aligned reads. For each condition, two independent replicate cultures are shown. (**B**) Representative genomic locus showing changes of DNA methylation in DNMT1 KO cells expressing ectopic DNMT1. Genome browser plots showing absolute and differential DNA methylation levels. DNA methylation levels are plotted in 10 kb genomic windows and are the mean of n = 2 independent replicate cultures. PMDs and HMDs are indicated by the coloured boxes. Chromosomal co-ordinates shown are hg38 chr9:0–40,000,000. (**C, D**) Boxplots (left) and heatmaps (right) showing differential DNA methylation between DNMT1 KO and DNMT1 KO cells expressing either eGFP or DNMT1. (**C**) Comparison of PMDs and HMDs. (**D**) Comparison of H3K9me3-marked PMDs and hypermethylated PMDs. Boxplots show the mean values of two independent replicate cultures. Heatmaps show replicates separately. In plots, n = 558 domains for HMDs, 546 domains for PMDs, 210 domains for H3K9me3-marked PMDs and 60 for hypermethylated PMDs. In all plots: + eGFP = DNMT1 KO expressing eGFP and +DNMT1 = DNMT1 KO expressing DNMT1. P-values shown are from two-sided Wilcoxon rank sum tests. For boxplots, lines = median; box = 25th–75th percentile; whiskers = 1.5 × interquartile range from box. All p-values are from two-sided Wilcoxon rank sum tests.
(PDF)

**S9 Fig. PMD-associated genes are upregulated in DNMT1 KO cells. A**) Density histograms of log10 RPKM values for PMD and HMD associated genes in HCT116 (left) and DNMT1 KO cells (right). HMDs: n = 11,243 genes and PMDs: n = 57 genes.(**B**) Barplot showing the % of significantly differentially expressed genes in PMDs and HMDs (n = 771 upregulated and 1,046 downregulated genes). (**C, D**) Boxplots showing log fold changes for DNMT1 KO vs HCT116 cells for genes located in PMDs versus HMDs (**C**) and hypermethylated PMDs versus H3K9me3-marked PMDs (**D**). In all plots, expression values were calculated form n = 9 and 4 independent cultures of HCT116 and DNMT1 KO cells respectively. In (**C** and **D**), n = 16,912 genes for HMDs, 1,235 genes for PMDs, 596 genes for H3K9me3-marked PMDs and 14 genes for hypermethylated PMDs. P-values shown are from two-sided Fisher's exact tests (**B**) and Wilcoxon rank sum tests (**C, D**). For boxplots, lines = median; box = 25th–75th percentile; whiskers = 1.5 × interquartile range from box.
(PDF)

**S10 Fig. Uncropped Western blots.** Uncropped scans of the Western blots associated with this study (from S4B-S4C Fig). The corresponding figure number is indicated for each blot. Membranes were cut to probe GAPDH loading controls on the same blot as experimental antibodies. Different exposure times were used for the loading control and experimental antibody. Additional samples not relevant to this study were loaded on some gels and these are not shown in the main figure.
(PDF)

**S1 Tables. Supplementary Tables.** Supplementary Tables A to J including sequencing summary statistics and PCR primers used in this study.
(XLSX)

## Acknowledgments

The authors wish to thank G. Brein for experimental advice and the Edinburgh Clinical Research Facility Genetics Core and Edinburgh Genomics for conducting the high-throughput sequencing used in this study.

## Author contributions

**Conceptualization:** Ioannis Kafetzopoulos, Francesca Taglini, Duncan Sproul.

**Formal analysis:** Ioannis Kafetzopoulos, Francesca Taglini, Moira Pasquier, Duncan Sproul.

**Funding acquisition:** Duncan Sproul.

**Investigation:** Ioannis Kafetzopoulos, Francesca Taglini, Hazel Davidson-Smith, Christine J Rodger, Lucia Puchades Gimeno, Andrew A Malcolm.

**Methodology:** Christine J Rodger, Lucia Puchades Gimeno, Andrew A Malcolm.

**Supervision:** Duncan Sproul.

**Writing – original draft:** Ioannis Kafetzopoulos, Duncan Sproul.

**Writing – review & editing:** Ioannis Kafetzopoulos, Francesca Taglini, Duncan Sproul.

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
