## [Editor Report · Decision Letter 0]

11 Jun 2025

PGENETICS-D-25-00493

DNMT1 loss leads to hypermethylation of a subset of late replicating domains by DNMT3A

PLOS Genetics

Dear Dr. Sproul,

Thank you for submitting your manuscript to PLOS Genetics. After careful consideration, we feel that it has merit but does not fully meet PLOS Genetics's publication criteria as it currently stands. Therefore, we invite you to submit a revised version of the manuscript that addresses the points raised during the review process.

We note the 'planned revisions' in your cover letter. Please proceed with them and we will then be in a position to assess whether to return the manuscript to your original reviewers for their further evaluation.

Please submit your revised manuscript within 60 days Aug 10 2025 11:59PM. If you will need more time than this to complete your revisions, please reply to this message or contact the journal office at plosgenetics@plos.org. Please include the following items when submitting your revised manuscript:

We look forward to receiving your revised manuscript.

Kind regards,

John M. Greally, D.Med., Ph.D.

Section Editor

PLOS Genetics

John Greally

Section Editor

PLOS Genetics

Aimée Dudley

Editor-in-Chief

PLOS Genetics

Anne Goriely

Editor-in-Chief

PLOS Genetics

**Journal Requirements:**

https://journals.plos.org/plosgenetics/s/submission-guidelines#loc-parts-of-a-submission

5) In the online submission form, you indicated that your data will be public upon acceptance. Please note that, though access restrictions are acceptable now, your entire minimal dataset will need to be made freely accessible if your manuscript is accepted for publication. This policy applies to all data except where public deposition would breach compliance with the protocol approved by your research ethics board.

2) If any authors received a salary from any of your funders, please state which authors and which funders.

7) Please ensure that the funders and grant numbers match between the Financial Disclosure field and the Funding Information tab in your submission form. Note that the funders must be provided in the same order in both places as well. Currently, the order of the grants is different in both places. In addition, this grant "18736" is missing from the  Financial Disclosure field.

**Figure resubmission:**
---

## [Decision Letter · Decision Letter 1]

16 Mar 2026

Dear Dr Sproul,

We are pleased to inform you that your manuscript entitled "DNMT1 loss leads to hypermethylation of a subset of late replicating domains by DNMT3A" has been editorially accepted for publication in PLOS Genetics. Congratulations!

In addition, Reviewer 2 had a few comments that you may choose to address as you perform the final formatting. No further review of these changes will be needed, the manuscript is accepted, it is your choice whether to make these minor modifications..

Yours sincerely,

John M. Greally, D.Med., Ph.D.

Section Editor

PLOS Genetics

John Greally

Section Editor

PLOS Genetics

Aimée Dudley

Editor-in-Chief

PLOS Genetics

Anne Goriely

Editor-in-Chief

PLOS Genetics

BlueSky: @plos.bsky.social

Comments from the reviewers (if applicable):

Reviewer's Responses to Questions

**Comments to the Authors:**

Reviewer #1: While the authors did not address most of my points with further experiments, they incorporated new analysis and suitable written explanations, some of which are incorporated in the revised manuscript. I have no further comments, and have no reservations about the work being published in its current form.

Reviewer #2: The authors have endeavoured to address most reviewers' comments, and the added analyses and experiments further the understanding of this model. In particular, the discussion of the role for H3K9me3 and H3K27me3 in the exploration of features of hyperPMDs in the results section is more balanced. I also appreciate the DNMT1 degron experiments are a substantial addition to demonstrate that changes in the DNMT1 KO are likely at least partially a consequence of differing disequilibriums between maintenance versus de novo DNA methylation rates across genomic regions.

It remains puzzling that DNMT1 KO hyperPMD domains exhibit such dramatic changes in DNA methylation and H3K9me3/H3K27me3/H3K36me2 without becoming Bonafide early replicating ‘active domains.’ The added analysis of gene upregulation in the DNMT1 KO hyper PMDs is a valuable addition for showing that only subtle gene upregulation is associated with these epigenetic changes. Potentially, ChIP-seq for active marks H3K4me3 and H3K27ac would have aided in this interpretation, as these would not be confounded by the low expression of genes within these domains as in RNA-seq.

The added comparison between HCT116 cells and primary colorectal cancers is also an informative addition, but it really does highlight that although there are distinctive dynamics of DNA methylation loss at hyperPMDs in colorectal cancers, the dramatic gains of methylation seen in the DNMT1 KO HCT116 cells is likely, at least in part, a consequence of selection in cell culture.

Overall, the detailed mechanistic interrogation of DNA methylation changes in a colorectal cell line provided in this study is informative in understanding the interplay between maintenance/de novo DNMTs, histone landscape, replication timing and gene expression.

Minor comment:

1. Discussion: While the results are more balanced in explaining the H3K9me3 and H3K27me3 loss at hyperPMDs, the discussion is very much still focused on the H3K9me3 finding. To reflect the results, I would suggest adding that H3K27me3 is also lost at hyperPMDs (top line, page 16 and top line of middle paragraph, page 17).

**Have all data underlying the figures and results presented in the manuscript been provided?**

Large-scale datasets should be made available via a public repository as described in the *PLOS Genetics*
data availability policy, and numerical data that underlies graphs or summary statistics should be provided in spreadsheet form as supporting information., and numerical data that underlies graphs or summary statistics should be provided in spreadsheet form as supporting information., and numerical data that underlies graphs or summary statistics should be provided in spreadsheet form as supporting information., and numerical data that underlies graphs or summary statistics should be provided in spreadsheet form as supporting information.

Reviewer #1: Yes

Reviewer #2: Yes

PLOS authors have the option to publish the peer review history of their article (what does this mean?). If published, this will include your full peer review and any attached files.). If published, this will include your full peer review and any attached files.). If published, this will include your full peer review and any attached files.). If published, this will include your full peer review and any attached files.

...

Reviewer #1: **Yes:**Maxim GreenbergMaxim GreenbergMaxim GreenbergMaxim Greenberg

Reviewer #2: No

**Data Deposition**

If you have submitted a Research Article or Front Matter that has associated data that are not suitable for deposition in a subject-specific public repository (such as GenBank or ArrayExpress), one way to make that data available is to deposit it in the Dryad Digital Repository. As you may recall, we ask all authors to agree to make data available; this is one way to achieve that. A full list of recommended repositories can be found on our . As you may recall, we ask all authors to agree to make data available; this is one way to achieve that. A full list of recommended repositories can be found on our . As you may recall, we ask all authors to agree to make data available; this is one way to achieve that. A full list of recommended repositories can be found on our . As you may recall, we ask all authors to agree to make data available; this is one way to achieve that. A full list of recommended repositories can be found on our website....

http://datadryad.org/submit?journalID=pgenetics&manu=PGENETICS-D-25-00493R1

Additionally, please be aware that our data availability policy requires that all numerical data underlying display items are included with the submission, and you will need to provide this before we can formally accept your manuscript, if not already present. requires that all numerical data underlying display items are included with the submission, and you will need to provide this before we can formally accept your manuscript, if not already present. requires that all numerical data underlying display items are included with the submission, and you will need to provide this before we can formally accept your manuscript, if not already present. requires that all numerical data underlying display items are included with the submission, and you will need to provide this before we can formally accept your manuscript, if not already present.

**Press Queries**

If you or your institution will be preparing press materials for this manuscript, or if you need to know your paper's publication date for media purposes, please inform the journal staff as soon as possible so that your submission can be scheduled accordingly. Your manuscript will remain under a strict press embargo until the publication date and time. This means an early version of your manuscript will not be published ahead of your final version. PLOS Genetics may also choose to issue a press release for your article. If there's anything the journal should know or you'd like more information, please get in touch via plosgenetics@plos.org....

---

## [Editor Report · Acceptance letter]

PGENETICS-D-25-00493R1

DNMT1 loss leads to hypermethylation of a subset of late replicating domains by DNMT3A

Dear Dr Sproul,

We are pleased to inform you that your manuscript entitled "DNMT1 loss leads to hypermethylation of a subset of late replicating domains by DNMT3A" has been formally accepted for publication in PLOS Genetics! Your manuscript is now with our production department and you will be notified of the publication date in due course.

With kind regards,

Anita Estes

PLOS Genetics

On behalf of:
